# Recycling Pretrained Checkpoints: Orthogonal Growth of Mixture-of-Experts for Efficient Large Language Model Pre-Training

## Abstract

The rapidly increasing computational cost of pretraining Large Language Models necessitates more efficient approaches. Numerous computational costs have been invested in existing well-trained checkpoints, but many of them remain underutilized due to engineering constraints or limited model capacity. To efficiently reuse this "sunk" cost, we propose to recycle pretrained checkpoints by expanding their parameter counts and continuing training. We propose orthogonal growth method well-suited for converged Mixture-of-Experts model: interpositional layer copying for depth growth and expert duplication with injected noise for width growth. To determine the optimal timing for such growth across checkpoints sequences, we perform comprehensive scaling experiments revealing that the final accuracy has a strong positive correlation with the amount of sunk cost, indicating that greater prior investment leads to better performance. We scale our approach to models with 70B parameters and over 1T training tokens, achieving 10.66% accuracy gain over training from scratch under the same additional compute budget. Our checkpoint recycling approach establishes a foundation for economically efficient large language model pretraining.

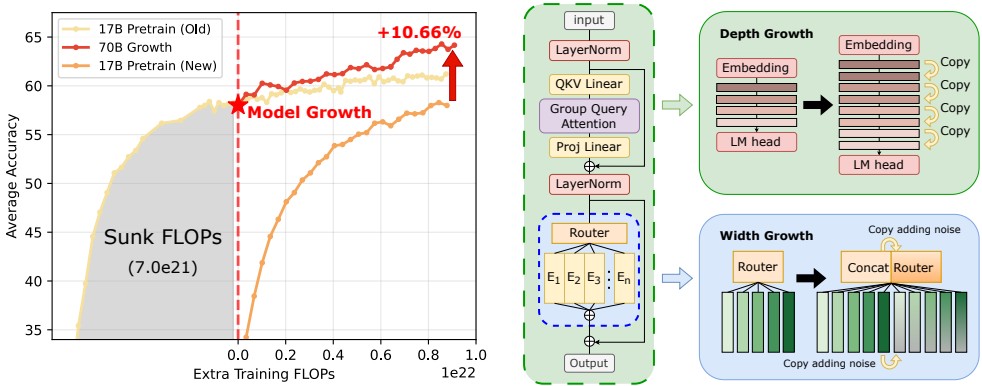

Figure 1: Main effect and method of our model growth framework

## 1 Introduction

The unprecedented success of large language models (LLMs) has been largely attributed to scaling laws (Kaplan et al., 2020; Hoffmann et al., 2022), which suggest that increasing model size and training data consistently improves performance. However, training these models from scratch demands enormous computational resources, and the exponential growth of this cost poses a fundamental barrier to further progress. Consequently, developing methods to scale models efficiently under constrained computational budgets has become a critical research challenge.

Modern LLM development pipelines routinely produce smaller pre-trained model checkpoints and numerous intermediate artifacts from processes like hyperparameter tuning or preliminary evalua-

tions. These models are often discarded once training concludes, leaving much of their potential unrealized due to inherent size constraints. We propose that these checkpoints represent a massive "sunk cost"—a significant computational investment that can be systematically leveraged. Model growth offers a new perspective on scaling: rather than starting from scratch, larger models can be created by "recycling" smaller pre-trained models, thereby inheriting their learned knowledge and optimized parameters.

However, recent studies on model growth seldom investigate its application to fully converged models. Existing works (Shen et al., 2022; Du et al., 2024) typically grow models after only a brief initial training period, a scenario that fails to leverage significant sunk costs. This work addresses a more pressing question: what is the optimal method for growing a well-trained model to maximize the return on its substantial sunk cost? Besides, with the increasing adoption of Mixture-of-Experts (MoE) architectures, it is crucial to investigate the effect of model growth on such structures, but to the best of our knowledge, this topic has not been systematically studied until now.

To address this gap, we develop a framework specifically for well-converged MoE models, proposing two orthogonal growth strategies: depth-wise expansion (adding layers) and width-wise expansion (increasing the number of experts), as illustrated in fig. 1 (right). We challenge the widely adopted "stacking" method for layer copying (Du et al., 2024; Wu et al., 2024), hypothesizing that it is suboptimal for converged models. Instead, we propose an "interpositional" method that better preserves the learned structural properties of the model, such as the characteristic trend in layer-wise weight norms. Moreover, we discover that adding a small amount of noise to newly copied experts is crucial as it facilitates better expert specialization.

We also provide a comprehensive study on the optimal timing for growth to best utilize the sunk cost. Our findings reveal a strong positive correlation between the amount of pre-training (measured in sunk FLOPs) and the final performance of the grown model. This confirms that a greater initial investment leads to a better final model, highlighting the efficacy of our framework in recycling prior computation. We further demonstrate that under a fixed total training budget (sunk + additional FLOPs), model growth is comparable or even slightly superior to training a large model from scratch.

Finally, we conduct extensive experiments to demonstrate the scalability and robustness of our orthogonal growth framework. As shown in fig. 1 (left), our method effectively scales an MoE model from 17 billion to 70 billion parameters using a 1-trillion-token dataset. The resulting model achieves a 10.66% average accuracy improvement on downstream tasks compared to a model trained from scratch with the same additional FLOPs budget.

In summary, our primary contributions are:

- We identify the **interposition** method as superior to **stacking** method for depth-growing **converged** models, as it better preserves the model's learned internal structure. We also introduce an optimized strategy for MoE width growth, showing that injecting Gaussian noise into new experts is critical for promoting effective specialization.

- We provide a comprehensive study on the optimal timing for model growth. We establish a **strong positive correlation** between the **sunk cost** (prior computation) of a base model and the final performance of the grown model.

- We validate the scalability of our framework by growing a 17B MoE model into a high-performing 70B model, which achieves a 10.66% accuracy gain over a scratch-trained baseline under the same extra FLOPs budget.

## 2 RELATED WORK

**Efficient Pretraining**. One direct approach for efficient model pretraining cost usage is to reduce computational costs, like model quantization (Jacob et al., 2018; Micikevicius et al., 2017; Peng et al., 2023; Wang et al., 2025), model pruning (Zhu & Gupta, 2017; Xia et al., 2022; Ma et al., 2023), and distillation (Gou et al., 2021; Loureiro et al., 2021; Sreenivas et al., 2024). An alternative approach focuses on reusing sunk cost to reduce the final training cost of the large model, like model growth (Shen et al., 2022) and upcycling (Komatsuzaki et al., 2023; Liew et al., 2025).

**Model Growth for Pretraining**. Model growth, or model expansion, is a technique to increase the number of parameters of pre-trained models or within the training process. Previous works such as Net2Net (Chen et al., 2015) focus on CNN models, while Bert2Bert (Chen et al., 2022), StackedBert (Gong et al., 2019), and MSG (Yao et al., 2024) have explored model growth techniques for BERT models. LEMON (Wang et al., 2024) and LiGO (Wang et al., 2023) further extend these approaches to other architectures such as vision transformers and DeiT. For Transformer-based architectures, studies such as Shen et al. (2022), Du et al. (2024), and Wang et al. (2024) investigate optimal growth strategies and initialization techniques, but these works are limited to relatively small models that are not trained on large-scale datasets. In the context of Large Language Models (LLMs), LLaMA Pro (Wu et al., 2024) proposes expanding the pre-trained LLaMA2-7B model to 8.3B parameters and fine-tuning it on new corpora, thereby improving knowledge coverage while mitigating catastrophic forgetting. Technical reports on Solar 10.7B (Kim et al., 2024) and FLM-101B (Li et al., 2023) also describe the adoption of model growth in large-scale pretraining, though details of the techniques and analyses are limited. Our work further extends the model growth paradigm to Mixture-of-Experts (MoE) architectures, and more importantly, to well-converged models. We have provided a comprehensive study on the definition for "converged" in section 3.1.

**Mixture-of-Experts Model Upcycling**. Mixture-of-Experts (MoE) (Shazeer et al., 2017; Zhou et al., 2022; Mu & Lin, 2025) is a classic model architecture widely adopted in large-scale models such as DeepSeek, Qwen-3, and LLaMA-4. Unlike the traditional Transformer architecture, MoE expands the Multi-Layer Perceptron (MLP) layers into multiple experts but activates only a subset during training. This design increases the overall model capacity while keeping the computational cost manageable. In contrast, traditional Transformer models without such sparsity are referred to as dense models. Recent works propose to initialize MoE models with existing dense checkpoints such as Sparse Upcycling (Komatsuzaki et al., 2023), thus reusing the sunk cost. Nakamura et al. (2025) and He et al. (2024) further explore this approach by introducing randomness or modifying expert granularity when transforming dense MLP layers into expert layers. Several technical reports, including Qwen-2 (Team, 2024) and Skywork-MoE (Wei et al., 2024), adopt this strategy to train MoE models from dense checkpoints. We extend this line of work by expanding existing MoE models into larger ones, which differs from the Upcycling approach in a key aspect: whether a pre-trained router is present. In the Upcycling setting, a new router must be randomly initialized, so prior works (Komatsuzaki et al., 2023; Muennighoff et al., 2024; Nakamura et al., 2025) need to add a large amount of noise (50% or more) to the new experts to encourage expert divergence. In our method, only a small amount of noise is needed for the copied experts and router to introduce divergence, and adding too much noise may become harmful.

# 3 GROWTH METHOD

This section introduces orthogonal growth strategies for Mixture-of-Experts (MoE) models. In section 3.1, we introduce **Depth Growth**, a method for expanding a model by duplicating its layers. In section 3.2, we present **Width Growth**, which involves expanding the number of experts. Finally, in section 3.3, we compare these two strategies and outline their respective advantages.

## 3.1 DEPTH GROWTH

Large Language Models (LLMs) are typically constructed from multiple transformer layers. Given a model $m$ with layers $l_1, l_2, \ldots, l_n$, a common method for layer-wise growth is called is **stacking**, which involves concatenating the original model's layers sequentially $k$ times:

$$M = \text{stack}(m) = \underbrace{l_1, l_2, \ldots, l_n, \; l_1, l_2, \ldots, l_n, \; \cdots, \; l_1, l_2, \ldots, l_n}_{k \text{ times}} \tag{1}$$

Alternatively, the **interposition** method duplicates each layer k times in place:

$$M = \text{interposition}(m) = \underbrace{l_1, l_1, \ldots, l_1}_{k \text{ times}}, \; \underbrace{l_2, l_2, \ldots, l_2}_{k \text{ times}}, \; \cdots, \; \underbrace{l_n, l_n, \ldots, l_n}_{k \text{ times}} \tag{2}$$

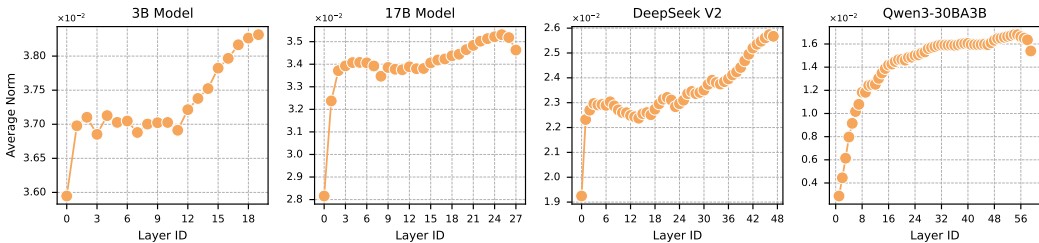

Figure 2: Characteristic layer-wise weight norm distribution in pre-trained LLMs, including pre-trained models in this work and from open-source community.

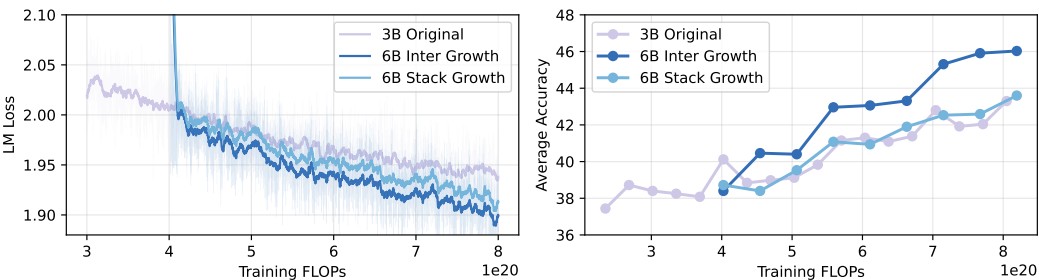

Figure 3: Performance comparison of interposition and stack depth growth strategies. Left: training loss; Right: average downstream task accuracy.

Previous studies, such as Wu et al. (2024) and Du et al. (2024), empirically advocate for the stack method. However, their work primarily focuses on the early stages of model training, before the parameter distributions of different layers have significantly diverged. We will demonstrate that for well-converged checkpoints, where layers have specialized roles, the stack method can be harmful to final performance. Furthermore, we provide quantitative experiments on this specific threshold.

As shown in fig. 2, the layer-wise weight norms of pre-trained models exhibit a distinct pattern: the norms of the initial layers are small and variable, followed by a gradual increase across the middle layers, and a slight decrease in the final layers. This trend is observable across several popular open-source models (see fig. 2) and we hypothesize that it is a signature of a healthy, stable pre-trained LLM. When grown from such converged checkpoints, clearly we should strive to maintain this upward trend in the norm as much as possible. So we hypothesize that the "stack" method disrupts this learned, position-dependent functional structure, whereas the "interposition" method preserves it, leading to better performance post-growth. We provide more examples in appendix B to further validate this observation. We will further show that this norm trend is actually related to the converged status of the model in training.

To conduct an end-to-end study, we trained a 3B-parameter MoE model with 20 layers and 64 experts from scratch. Four experts are activated during computation, yielding a total of 500M activated parameters. This model was then grown to 6B parameters (with 800M parameters activated) to evaluate the effects of each growth strategy. In our experiments, the growth factor $k$ in eq. (2) is fixed to 2 following peer work (Shen et al., 2022; Wang et al., 2024). Further details regarding model pre-training are available in appendix D. Figure 3 shows the results. To ensure a fair comparison given the increased size of the grown models, the x-axis represents the total training Floating Point Operations (FLOPs). Based on both training loss and average downstream task accuracy[1], the interposition method outperforms the stack method. We also obtain similar results on the larger 17B model in our experiments (see fig. 11 in section 5).

In order to quantitatively define the boundary condition of the superiority of interposition method, we conducted a systematic study using the **amount of training FLOPs** as a measure of training progress. According to the Chinchilla Scaling Law (Hoffmann et al., 2022), the compute-optimal ratio between model size (N) and token count (D) is approximately $D = 20N$, and the corresponding

---

[1]The accuracy metric is computed as the average score across multiple downstream evaluation tasks, such as MMLU, ARC-C, HellaSwag, BoolQ and OpenbookQA. The computation method is detailed in appendix E.1, and full results are in appendix E.2.

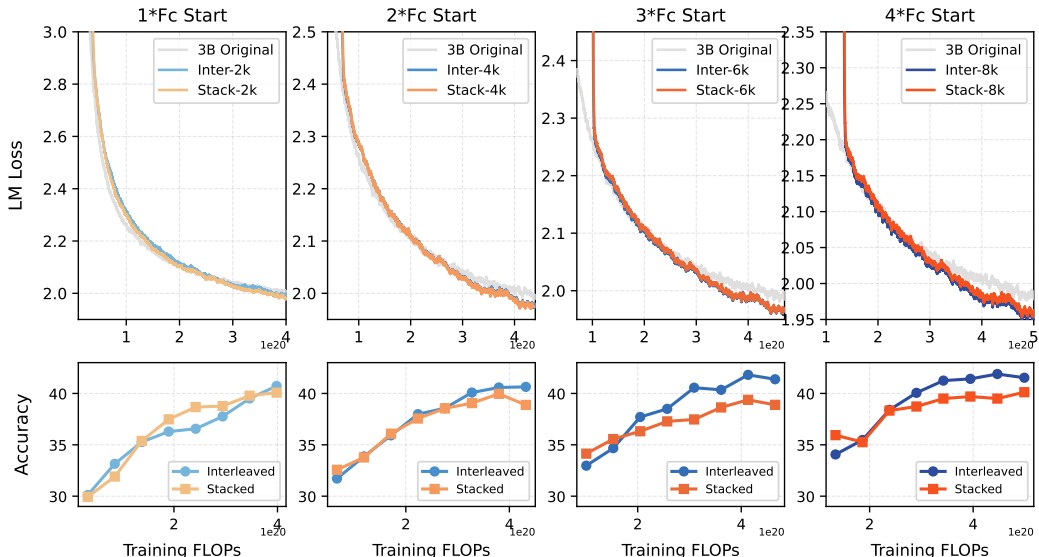

Figure 4: Investigation on the boundary condition for interposition method's superiority over stack method, provided with loss curves and accuracy evaluation results based on different growth timing.

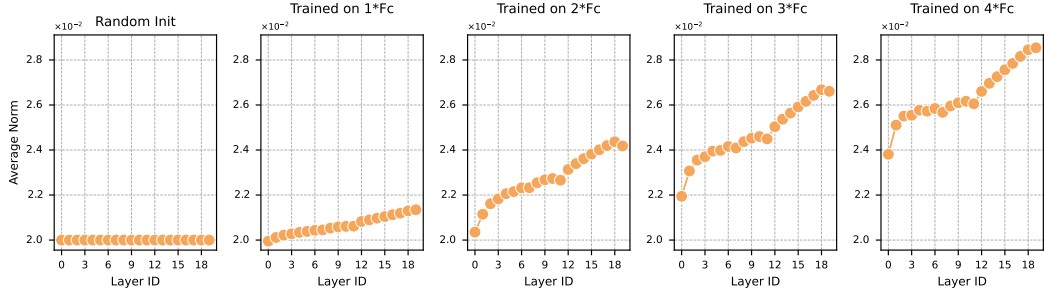

Figure 5: Investigation on the variation trend for average weight norm during model training process.

total compute can be estimated as $6ND$ (where the factor 6 follows from the empirical rule of forward + backward computation). For MoE models, the situation is more complicated, but a reliable approximation is to base the compute on the number of activated parameters ($N_a = 500M$) instead of the total parameter count (Fedus et al., 2022). For our 3B_A500M model, the resulting compute-optimal FLOPs ($F_c$) is approximately $F_c \approx 6 \cdot N_a \cdot (20N_a) \approx 3 \times 10^{19}$. In practice, small models are commonly overtrained well beyond this compute-optimal value.

Using the accumulated FLOPs of the 3B_A500M model as the indicator, we examined checkpoints at various training stages. At each selected checkpoint, we applied either stack or interpositional layer growth, then continued training for a fixed number of steps and compared their performance. We selected several checkpoints roughly corresponding to integer multiples of $F_c$ as growth points.

Based on the findings in fig. 4, we observe that the critical point at which the two growth methods begin to diverge emerges at approximately **2× the Chinchilla-optimal FLOPs (($F_c$))**. This suggests that once the total training FLOPs exceed twice $F_c$, the interpositional method should be preferred over the stack method, indicating that the model has reached a well-converged state.

To further connect this observation with our findings on weight norms, fig. 5 shows the weight norm distributions of the checkpoints used in this experiment. Early in training, all weights are initialized from an i.i.d. Gaussian distribution with a fixed standard deviation ($\delta = 0.02$), leading to uniform layer-wise norms. As training progresses, the weight-norm distribution develops a clear upward trend. Around 4k steps ($2F_c$), a stable layer-wise increasing pattern emerges, and in subsequent training the norms continue to grow while maintaining this structure. Therefore, we believe that **the emergence of this characteristic increasing pattern across layers marks the boundary at which interpositional growth begins to outperform stack growth**. In conclusion, for converged models

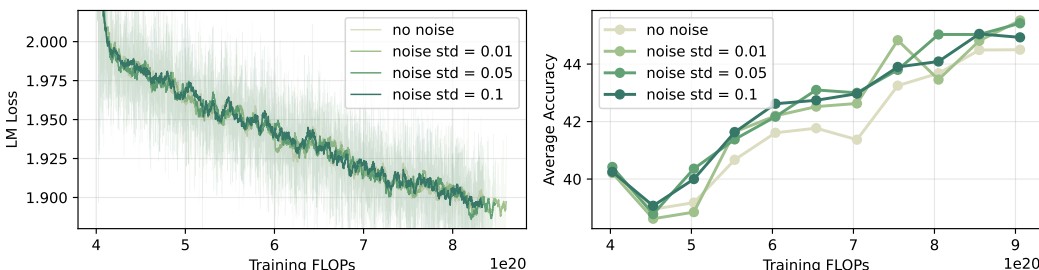

Figure 6: The impact of noise injection scale on width growth performance. Left: training loss; Right: average downstream task accuracy.

rather than those in the early stages of training, the interpositional method is a better choice than the widely adopted stacking method.

## 3.2 WIDTH GROWTH

For MoE models, an alternative to increasing depth is to expand the parameter count by increasing the number of experts. To preserve the capabilities of a converged MoE model during such growth, it is crucial to proportionally increase the number of activated experts (the top-k parameter) as tokens must be routed to the newly added capacity.

In our experiments, we simultaneously double both the total number of experts and the number of activated experts (the value of $k$). For an original MoE layer with $E$ experts and a top-k routing scheme, the output is given by:

$$\text{MoE}(x) = \sum_i g_i(x)\, f_i(x), \quad i \in \text{Top}_k\big(g(x)\big), \quad g(x) \in \mathbb{R}^E \tag{3}$$

Where $f_i$ is the $i$-th expert and $g(x)$ is the vector of gating weights from the router. When the number of experts is doubled to $2E$ and the number of activated experts to $2k$, the formulation becomes:

$$\text{MoE}_{\text{growth}}(x) = \sum_i g_i'(x)\, f_i'(x), \quad i \in \text{Top}_{2k}\big(g'(x)\big), \quad g'(x) \in \mathbb{R}^{2E} \tag{4}$$

A critical aspect of MoE training is achieving both load balancing and expert specialization, which ensures that tokens are distributed evenly across experts and that different experts learn distinct functions. In our growth scenario, we first duplicate each expert to preserve the model's learned capabilities. To encourage the new experts to diverge and learn new knowledge, we propose adding Gaussian noise to the weights of the newly created experts and to the corresponding logits in the router. Specifically, we add noise with a mean of 0 and a standard deviation of $\alpha \times \sigma_{\text{orig}}$, where $\sigma_{\text{orig}}$ is the standard deviation of the original weights. The new expert and router weights are then concatenated with the original ones. To promote divergence without destabilizing the well-trained original experts, we use a small value for $\alpha$ number such as $\alpha = 0.01$.

As the experimental results in fig. 6 reveal, while the language modeling loss is similar for both direct expert copying ($\alpha = 0$) and the noise-addition method, the latter demonstrates better performance on downstream tasks, yielding an accuracy improvement of approximately 1%. The results also indicate that excessive noise may be harmful. These findings validate the importance of adding a small magnitude of noise to stimulate expert specialization during width growth.

## 3.3 DISCUSSION ON DEPTH AND WIDTH GROWTH

Increasing a model's depth and width are two orthogonal growth strategies for MoE models. In our study, we investigate the distinct characteristics of these two methods. For general downstream task performance, fig. 7 shows that depth growth generally yields better results than width growth. Width growth requires more continued training for the expanded set of experts to achieve a balanced load distribution and specialize effectively. Thus, its benefits are less immediate than those of depth growth.

However, width growth holds advantage in preserving model stability, because it aligns well with the principle of **Function-Preserving Transformations** (Evci et al., 2022; Wang et al., 2023; Yao et al., 2024), which stipulate that a model's output should remain unchanged immediately after expansion. Further discussion and a proof regarding this property are provided in appendix C.

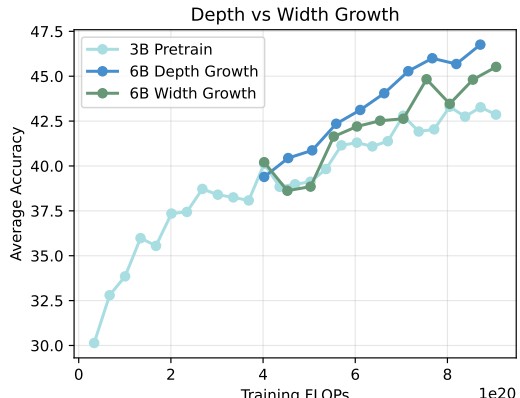

Figure 7: Comparative analysis of performance between depth and width growth.

# 4 ANALYSIS OF GROWTH TIMING AND SUNK COST

Having established the efficacy of our growth methods, we now turn to a critical practical question: when is the optimal time to apply them? In this section, we investigate the optimal point during the pre-training process to apply the growth strategy and compare its efficacy against training a larger model from scratch. We demonstrate that even for already converged trained checkpoints, model growth can still effectively leverage the computational investment (i.e. sunk FLOPs cost).

## 4.1 IMPACT OF SUNK COST WITH A FIXED ADDITIONAL BUDGET

This analysis addresses a primary questions: given a series of checkpoints with varying amounts of sunk cost, which serves as the optimal base for growth? Specifically, does a greater sunk cost lead to superior performance post-growth?

Expanding on the experiments in section 3, we trained the 3B_A500M MoE model to full convergence using a standard learning rate schedule, which included warmup, a constant learning rate phase, and an annealing phase (see fig. 8). We saved a series of checkpoints throughout this process, each representing a different level of sunk cost. To evaluate the benefit of this investment, we conducted experiments with a fixed budget for additional training FLOPs. Since depth growth ultimately yields better results than width growth, we focus exclusively on depth growth in these experiments. We selected 12 checkpoints, sampled between 8k and 96k training steps, and grew each to 6B parameters. We also include a baseline where a 6B_A800M model is trained from scratch, which is equivalent to growing a model with zero sunk FLOPs.

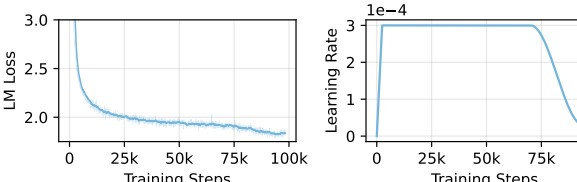

Figure 8: Full training curve and learning rate scheduler of 3B_A500M model pretraining.

The results of growing models from different checkpoints, each with the same budget for additional FLOPs, are shown in fig. 9. Both the final training loss and the average downstream accuracy exhibit a strong positive correlation with the sunk cost invested prior to growth. This indicates that a larger initial training investment leads to a better final model, confirming that the growth method effectively recycles prior computational work, and provide suggestion that latter checkpoints with more sunk cost can be leveraged to get better growth performance. We further present quantitative results in table 1 to support this positive correlation. The table reports the starting, ending, and average accuracy across the entire continued training process with an additional $3 \times 10^{20}$ FLOPs.

Notably, while the positive correlation persists when the base model enters the learning rate annealing stage (beyond 72k steps), the marginal performance gains diminish. This is likely because all grown models in this experiment are trained with the same new constant learning rate for fair comparison, which may not be optimal for a checkpoint from a late annealing phase. This suggests

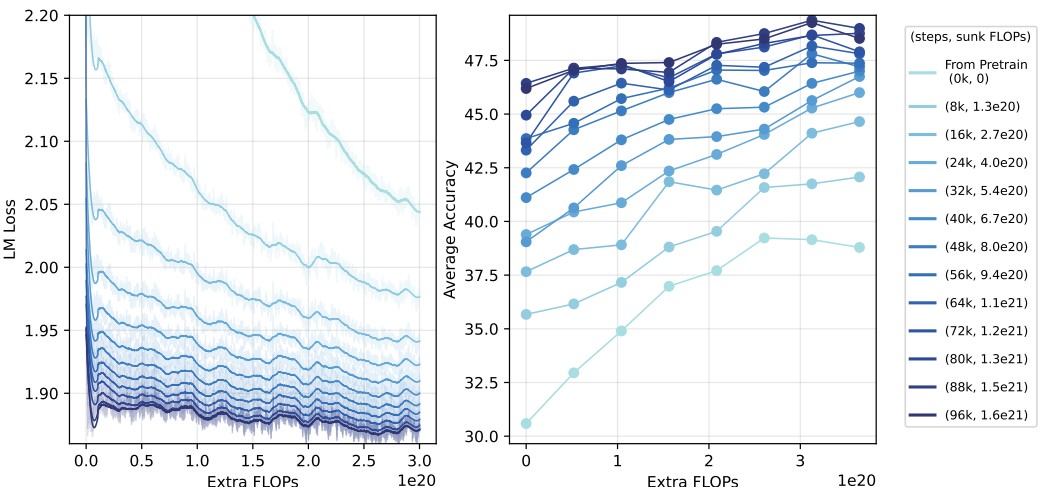

Figure 9: Investigation of growth time according to amount of sunk cost. Left: loss curve. Right: downstream tasks average accuracy.

Table 1: Quantitative accuracy results growth time investigation for amount of sunk cost.

| Start steps | 0k | 8k | 16k | 24k | 32k | 40k | 48k | 56k | 64k | 72k | 80k | 88k | 96k |
|---|---|---|---|---|---|---|---|---|---|---|---|---|---|
| Start acc | 30.59 | 35.67 | 37.66 | 39.39 | 39.05 | 41.11 | 42.26 | 43.86 | 43.32 | 43.66 | 44.95 | 46.43 | 46.19 |
| End acc | 38.79 | 42.07 | 44.65 | 46.00 | 46.75 | 47.00 | 47.20 | 47.37 | 47.81 | 47.90 | 48.76 | 48.99 | 48.52 |
| **Average acc** | 36.29 | 39.09 | 41.19 | 42.69 | 43.34 | 44.51 | 45.67 | 46.15 | 46.49 | 47.13 | 47.43 | 47.88 | 47.82 |

that one should either carefully tune the learning rate for the continued training phase or, preferably, select a checkpoint from the constant learning rate phase for growth.

## 4.2 COMPARISON TO SCRATCH TRAINING WITH A FIXED TOTAL BUDGET

The results in fig. 9 also demonstrate that, for a fixed *additional* training budget, model growth is clearly superior to training from scratch. We next investigate whether this advantage holds when the *total* FLOPs budget is fixed. The results of this experiment are presented in fig. 10. Here, models grown from later checkpoints are allocated a correspondingly smaller budget for continued training.

The results show that for most growth timings, the final accuracy of the grown model is comparable or slightly superior to the scratch-trained model. Specifically, models grown from earlier checkpoints, which thus allocated a larger proportion of the total budget for post-growth training, tend to perform best. This suggests that the pre-trained smaller model serves as a highly effective initialization for the larger model's training process. The growth method underperforms only when initiated from a very late checkpoint, where the budget for continued training is insufficient. This provides a valuable heuristic: one should allocate additional FLOPs at least on the same order of magnitude as the sunk cost in order to achieve performance comparable to pre-training under the same total FLOPs.

Quantitative results are provided in table 2 to support this finding, showing the average accuracy over the final six accuracy measurements (with the exception of line 64k, which contains only four data points). Notably, although the training loss of later checkpoints remains relatively high, the final accuracy quickly recovers during continued training.

In conclusion, model growth is an effective strategy for leveraging the sunk cost of pre-trained models, with final performance positively correlating with the initial training investment. Furthermore, its effectiveness is comparable and sometimes superior to training from scratch, even when evaluated under a fixed total-FLOPs budget.

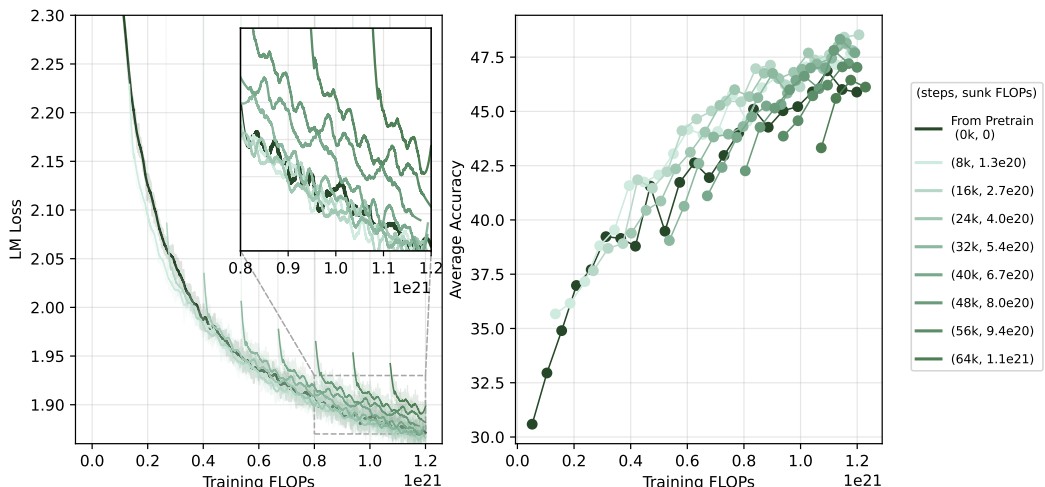

Figure 10: Investigation of growth time according to total amount of training FLOPs. Left: loss curve. Right: downstream tasks average accuracy.

Table 2: Quantitative accuracy results growth time investigation for total FLOPs.

| Start steps | 0k | 8k | 16k | 24k | 32k | 40k | 48k | 56k | 64k |
|---|---|---|---|---|---|---|---|---|---|
| End acc | 45.03 | 47.66 | 48.53 | 47.80 | 48.15 | 47.70 | 47.20 | 47.03 | 46.12 |
| **Average acc** | 45.82 | 46.99 | **47.38** | 47.29 | 47.05 | 46.96 | 46.47 | 45.74 | 45.37 |

# 5 SCALABILITY EXPERIMENTS

The practical value of model growth depends on its scalability, since training larger models comes with proportionally higher sunk costs. To this end, we scale our experiments to a 17B-parameter MoE model, which we progressively grow to a 70B model over one trillion training tokens. This large-scale experiment demonstrates the robustness and effectiveness of our proposed methods.

We employ the same growth techniques introduced in section 3. As a preliminary step, we re-validate our findings on depth growth at the 17B

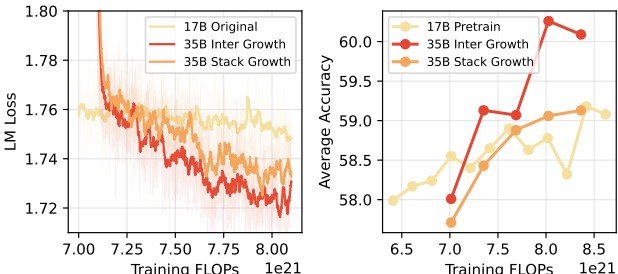

Figure 11: Performance comparison of interposition and stack depth growth strategies for 17B model. Left: training loss; Right: average downstream task accuracy.

scale. The 17B base model's architecture is a scaled-up version of the 3B model, with 1.76B parameters activated. The complete architectural and training details are available in Appendix D. The results, shown in fig. 11, confirm that the interposition method remains superior to the stack method, further substantiating our central insight regarding the growth of converged checkpoints.

In this scalability experiment, we first expand the model's depth to increase its functional capacity, then broaden its width to enhance expert specialization. This is also a good example to validate the independence and orthogonality of our proposed two growth method. First, we train the initial 17B model (with 4 activated experts) for approximately 600B tokens. At this point, we perform **Depth Growth**, increasing the number of layers from 28 to 54, which results in a 35B model (3.1B parameters activated). After training this intermediate model for an additional 300B tokens, we perform **Width Growth**, doubling the number of experts from 96 to 192. This yields the final 70B model

(5.2B parameters activated), which is then trained for another 100B tokens. The complete training loss curve and downstream evaluation results are presented in fig. 12 and fig. 13, respectively.

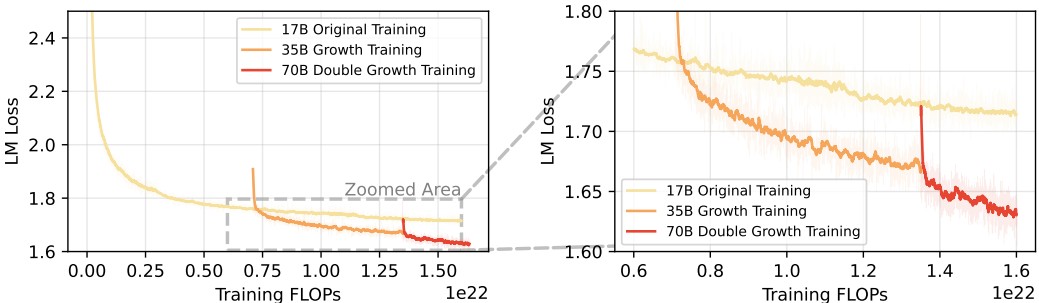

Figure 12: Full training loss for 17B_A1.76B model pretraining and growth training. Left: original loss curves. Right: zoom in for better visualization.

The experimental results reveal a critical finding: our growth method can unlock substantial performance gains even after the base model's improvement has saturated following extensive training. Furthermore, the sequential application of depth and width growth creates a well-proportioned final architecture, which leads to superior overall performance compared to the intermediate models. As shown in fig. 13, the final 70B_A5.2B model achieves an average accuracy of 64.17, representing a notable improvement of 2.21

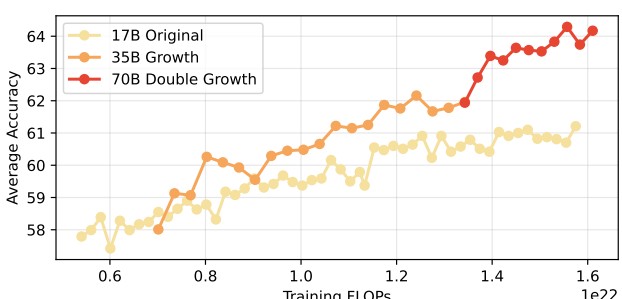

Figure 13: Downstream task evaluation result for 17B_A1.76B model pretraining and growth training.

points (61.96 → 64.17) over the 35B_A3.1B checkpoint and 5.62 points (58.55 → 64.17) over the initial 17B model. Even under the same training FLOPs, the 70B_A5.2B model outperforms the 17B_A1.76B model by 2.96 points (approximately 4.0% relative to 61.71). From the perspective of sunk cost utilization, the growth model also demonstrates superior performance, surpassing the model trained from scratch by 6.18 points (approximately 10.6% relative to 57.99) under the same extra FLOPs budget, as shown in fig. 1.

These large-scale results reaffirm that model growth is a powerful and efficient strategy for leveraging the computational investment in existing checkpoints while pushing the performance boundaries of the resulting model.

## 6    CONCLUSION

In this work, we propose a systematic framework for model growth, addressing the computational cost problem in large language model pretraining. We demonstrate that pre-trained checkpoints, often considered disposable assets, can be effectively "recycled" to create larger and more capable models, thus preserving their significant sunk cost. We identify optimal strategies for two orthogonal growth dimensions in Mixture-of-Experts (MoE) models, establish a scaling principle that growing from a more converged checkpoint yields superior final performance, and demonstrate that our framework is highly scalable. By redefining pre-trained checkpoints as valuable foundations for future growth, our methods contribute to a more sustainable and accessible path for pre-training Large Language Models.

ETHICS STATEMENT

This research focuses on developing methods for efficiently scaling large language models through model growth, with the primary goal of reducing computational costs and reusing previously trained checkpoints. The study does not involve human subjects, personal data, or sensitive demographic information. Potential societal impacts of large language models are acknowledged, such as misuse for generating harmful or biased content, but this work does not introduce new risks beyond those already inherent in the use of such models. Instead, by improving training efficiency, the proposed methods may lower the environmental footprint of model development.

REPRODUCIBILITY STATEMENT

We have taken several steps to ensure the reproducibility of our results. The main paper provides detailed descriptions of the proposed model growth framework and experimental settings, while additional implementation details, dataset composition, hyperparameters, and training configurations are included in the appendix D. To further facilitate reproducibility, we provide anonymized source code fragment in the supplementary materials, and we will release our training framework to facilitate further research in this area. Figures and tables referenced in the main text are generated directly from logged experimental outputs listed in appendix E.2, ensuring that reported results can be consistently verified.

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

## A   USE OF LARGE LANGUAGE MODELS

Large Language Models (LLMs) were used only to polish the writing (e.g., grammar, style, and readability). All research ideas, methods, experiments, and analyses were fully developed and conducted by the authors.

## B   MORE RESULTS ON LAYER-WISE NORM DISTRIBUTION

We further extend our analysis by examining a broader range of open-source MoE models. Specifically, we compute and visualize the layer-wise average weight norm distributions for Deepseek-v2-Lite-16B-A2.4B (DeepSeek-AI, 2024), Qwen1.5-MoE-14.3B-A2.7B-Chat (Yang et al., 2025), Mixtral-8x7B (Jiang et al., 2024), Hunyuan-A13B-Instruct (Sun et al., 2024), Dots-LLM1-142B-A14B (Huo et al., 2025), and GroveMoE-Inst-33B-A3.2B (Wu et al., 2025).

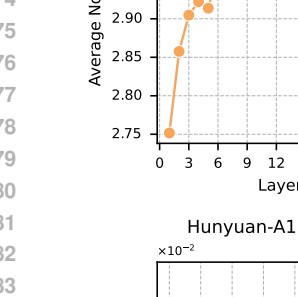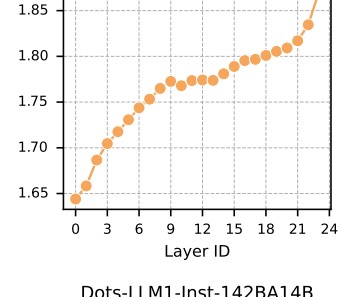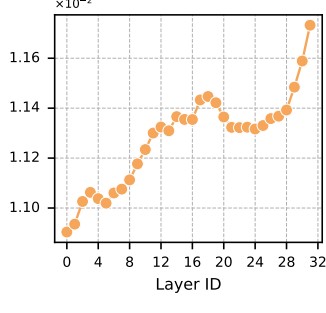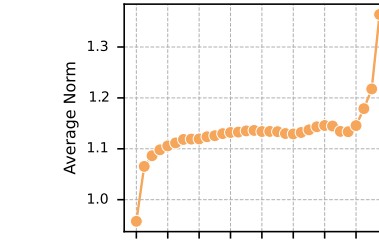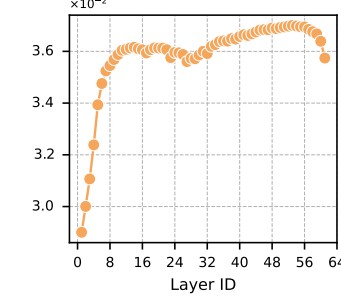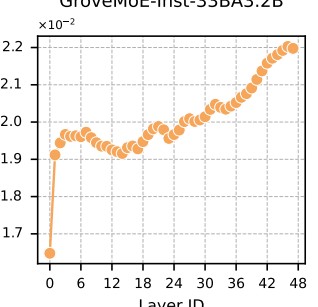

Figure 14: Characteristic layer-wise weight norm distribution in pre-trained LLMs from several open-source models.

As shown in fig. 14, a consistent pattern emerges across well-converged MoE models: the layer-wise weight norms tend to increase with depth. This trend provides further empirical support for our proposed interpositional growth method (section 3.1), highlighting its ability to align with the intrinsic training dynamics of large MoE architectures.

## C   DISCUSSION ON FUNCTION PRESERVING

We observed that, under our model architecture, directly growing a smaller model into a larger one does not lead to severe accuracy degradation on downstream evaluations, even though the outputs for identical inputs may differ due to manual alterations of model weights. Furthermore, the accuracy drop tends to be smaller for width growth compared to depth growth. This phenomenon relates to a principle in model growth known as **Function Preserving** (FP) (Evci et al., 2022; Wang et al., 2023; Yao et al., 2024). FP stipulates that, for any given input, the output before and after model growth should remain identical, thereby guaranteeing that performance is not immediately harmed: $y_{\text{original}}(x) = y_{\text{growth}}(x)$. In practice, however, we find that even when FP rules are not strictly

enforced, performance degradation is minor. This robustness can be attributed to the pre-norm structure widely adopted in modern transformers. In **pre-norm** layers, the normalization is applied *before* the residual connection, i.e.,

$$h^{(l+1)} = h^{(l)} + \mathcal{F}\big(\text{LN}(h^{(l)})\big), \tag{5}$$

where $h^{(l)}$ is the input to layer $l$, LN denotes layer normalization, and $\mathcal{F}$ represents the sublayer transformation (e.g., attention or feedforward block).

By contrast, in the original Transformer and BERT, the **post-norm** structure was used, where normalization is applied *after* the residual connection:

$$h^{(l+1)} = \text{LN}\big(h^{(l)} + \mathcal{F}(h^{(l)})\big). \tag{6}$$

Although post-norm structures can better exploit model capacity, they are known to be harder to optimize and less stable during training. Pre-norm designs, in contrast, are easier to train but may reduce the model's effective depth.

This structural distinction explains our empirical findings. Under the pre-norm structure, when layers are duplicated during depth growth, the residual-normalization combination in eq. (5) ensures that the difference between the output of a single layer and that of a duplicated pair of layers is small. As a result, the overall model output remains similar, and performance degradation is limited. In contrast, with post-norm (eq. (6)), duplicating layers alters the scale of normalized outputs more substantially, leading to larger deviations and thus greater performance drops immediately after growth.

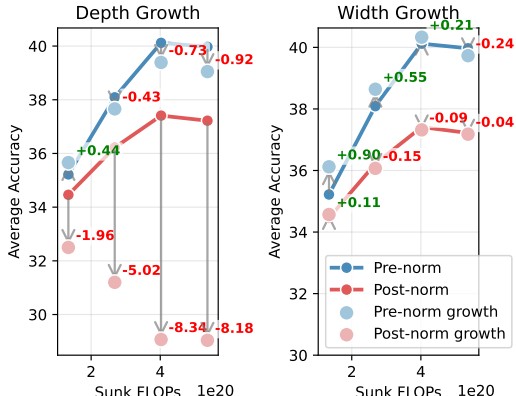

Experimental evidence supporting this claim is presented in fig. 15, where we find that evaluating a checkpoint immediately after width growth (before any further training) results in only a minor decrease in downstream task ac-

Figure 15: Experimental results on Function-Preserving principle between depth and width growth.

curacy, or in some cases even a slight improvement due to the inherent randomness of evaluation. In contrast, depth growth can disrupt the functional role of layers, and in older post-layer normalization (Post-LN) architectures like BERT and original Transformer, this would cause a significant performance degradation immediately after expansion. Width growth, however, effectively preserves performance post-expansion in both Pre-LN and Post-LN architectures.

From another perspective, width growth is naturally more function-preserving. When adding experts in MoE layers, both expert weights and router weights are copied. This implies that, at inference time, the widened MoE produces outputs identical to the original configuration, fully consistent with the FP principle. In practice, we add small Gaussian noise to the new experts to encourage specialization during continued training, but such noise only causes negligible shifts in model outputs. Importantly, since width growth operates solely within the MoE module and does not alter the layer structure, it maintains performance under both pre-norm and post-norm settings. This explains why width growth yields better immediate performance retention than depth growth, as shown in fig. 15.

## D    DETAILED TRAINING SETTINGS

This appendix provides details of our pretraining pipeline, including model architecture (appendix D.1), dataset composition (appendix D.2), training hyperparameters (appendix D.3), and infrastructure configurations (appendix D.4).

## D.1 Model Structure

We adopt a standard decoder-only LLM architecture, with each layer containing Grouped Query Attention (GQA) and Mixture-of-Experts (MoE) modules in both the 3B and 17B models. RMSNorm is used for layer normalization, and rotary position embeddings are applied.

For the 3B model, we set the number of layers to 20 with a hidden size of 1024. The GQA module uses 16 attention heads grouped into 4 query groups. The MoE module consists of 64 experts, of which 4 are activated during computation. The hidden size of each expert is 768.

For the 17B model, we use 28 layers with a hidden size of 2048. GQA again uses 16 attention heads with 4 query groups. The MoE module includes 96 experts, with 6 activated during computation. Each expert has a hidden size of 1024.

For detailed number of parameters and the activated number of parameters of original 3B/17B model and their growth variants: For the 3B model (3,330,979,840 parameters), 500M (498,595,840) parameters are activated excluding the embedding layer. After growth, the 6B-depth model (6,457,093,120) activates 792M (792,325,120) parameters, while the 6B-width model (6,352,189,440) activates 687M (687,421,440). For the 17B model (17,620,395,008), 1.76B (1,760,764,928) parameters are activated. After growth, the 35B-depth model (34,831,056,896) activates 3.1B (3,111,796,736) parameters, and the 35B-width model (34,537,333,760) activates 2.8B (2,818,073,600). The final 70B model (68,664,934,400) activates 5.2B (5,226,414,080).

For MoE models specifically, we apply a sigmoid function to compute router scores instead of the softmax function. In the router, expert bias is disabled for the 3B model but enabled for the 17B model. For load balancing, we use sequence-level auxiliary loss in the 3B model and global-batch auxiliary loss in the 17B model.

## D.2 Dataset Composition

Our pretraining corpus is constructed from a diverse and high-quality dataset comprising a mixture of public and proprietary sources, including:

- **DCLM**: dataset released by Apple (Li et al., 2024) with de-duplication (1T tokens)
- **FineWeb-Edu**: dataset released by Hugging Face (Penedo et al., 2024) with de-duplication (280B tokens)
- **Nemotron-CC-HQ**: a high-quality Common Crawl–based dataset (4.67T tokens) released by NVIDIA (Su et al., 2024)
- **Filtered Code Data**: a curated code dataset (640B tokens)
- **Synthetic Data**: high-quality, instruction-oriented synthetic corpora (1.8T tokens)

We randomly shuffle these corpora and uniformly sample approximately 1T tokens for training. We preprocess the raw dataset using the GPT-4o tokenizer, which has a vocabulary size of 200,019. The maximum sequence length is fixed at 4096 tokens. During training, the batch size is set to 1024 for the 3B model and 4096 for the 17B model.

## D.3 Training Hyperparameters

Both for 3B model and 17B model, all learnable parameters are randomly initialized with a standard deviation of 0.02. We employ the AdamW optimizer (Loshchilov et al., 2017) with hyper-parameters set to $\beta_1 = 0.9$, $\beta_2 = 0.95$, and weight-decay = 0.1. Max learning rate is $3 \times 10^{-4}$ for 3B model and $2.6 \times 10^{-4}$ for 17B model. As for the learning rate scheduling, we first linearly increase it from 0 to max learning rate during the first 3K steps. Then, we keep a constant learning rate. For 3B model, we decay it into the minimum leaning rate, which is 1/10 of max learning rate during the annealing process. We donot do annealing for 17B model yet.

## D.4 Infrastructure Details

We train our model with mixed precision framework (BF16 + FP32). We use Flash Attention(Dao et al., 2022) for training acceleration. A distributed optimizer is employed to partition optimizer

states across data-parallel GPUs, thereby reducing memory consumption. MoE layer recomputation is enabled to further decrease memory usage, and Grouped GeMM (General Matrix Multiplication) is used to accelerate MoE computations. For the 17B model, we use an expert parallel size of 8 to distribute expert weights across GPUs, which allows us to fit within the memory constraints of each device. For infrastructural reasons, we occasionally enable pipeline parallelism (size = 2) to free memory for larger microbatch sizes, improving GeMM efficiency. For the smaller 3B model, we use an expert parallel size of 2 without pipeline parallelism, since the cost of all-to-all expert communication is lower than the overhead introduced by pipeline scheduling and idle bubbles.

# E    EVALUATION DETAILS

## E.1    METHOD FOR COMPUTING AVERAGE ACCURACY

We conduct our evaluation through the widely used lm-evaluation-harness library[2] (Gao et al., 2024). All reported average accuracy values in the main text are derived from the average accuracy of following two categories: (1) comprehensive knowledge and reasoning ability, and (2) basic multiple-choice QA performance.

For comprehensive knowledge and reasoning ability, we use the MMLU benchmark (Massive Multi-task Language Understanding) (Hendrycks et al., 2020), which consists of 57 tasks spanning STEM, humanities, social sciences, and professional domains. MMLU is widely recognized for assessing models' ability to apply world knowledge, solve problems, and perform reasoning beyond surface-level pattern recognition. We evaluate using a few-shot setting with 5 in-context examples.

In addition, we assess performance on multiple-choice QA benchmarks including ARC (Clark et al., 2018), BoolQ (Clark et al., 2019), HellaSwag (Zellers et al., 2019), LogiQA (Liu et al., 2021), OpenBookQA (ObQA) (Mihaylov et al., 2018), and Winogrande (Sakaguchi et al., 2021). These tasks are evaluated in the zero-shot setting, with accuracy (percentage of correctly chosen options) as the evaluation metric. Collectively, they complement MMLU by emphasizing commonsense and scientific reasoning in narrower but challenging domains.

## E.2    DETAILED EVALUATION RESULTS

We provide the complete accuracy tables from table 3 to table 26. The results presented in the main text as averaged figures or tables are derived directly from these original tables. For clarity, we indicate the corresponding appearances of each result in the table headers.

---

[2]https://github.com/EleutherAI/lm-evaluation-harness

Table 3: Full evaluation results of 3B model pretraining in fig. 3 and fig. 7

| Steps | MMLU | QA_average | Arc_C | BoolQ | Hellaswag | Logiqa | Openbookqa | Winogrande | **Average** |
|---|---|---|---|---|---|---|---|---|---|
| 2k | 25.54 | 34.75 | 22.87 | 54.34 | 27.60 | 28.88 | 24.20 | 50.59 | 30.14 |
| 4k | 27.69 | 37.91 | 28.75 | 49.79 | 35.41 | 28.88 | 30.40 | 54.22 | 32.80 |
| 6k | 28.94 | 38.76 | 32.85 | 46.30 | 39.95 | 29.19 | 31.60 | 52.64 | 33.85 |
| 8k | 30.70 | 41.26 | 34.56 | 53.36 | 43.62 | 29.65 | 32.60 | 53.75 | 35.98 |
| 10k | 29.08 | 42.02 | 35.84 | 55.32 | 45.24 | 28.42 | 33.40 | 53.91 | 35.55 |
| 12k | 31.75 | 42.95 | 34.39 | 57.49 | 47.51 | 29.49 | 33.40 | 55.41 | 37.35 |
| 14k | 32.46 | 42.42 | 37.12 | 48.93 | 48.85 | 29.19 | 34.20 | 56.20 | 37.44 |
| 16k | 33.34 | 44.10 | 37.03 | 56.79 | 50.15 | 29.65 | 34.40 | 56.59 | 38.72 |
| 18k | 32.93 | 43.87 | 39.85 | 52.54 | 50.69 | 30.57 | 34.00 | 55.56 | 38.40 |
| 20k | 33.46 | 43.04 | 39.33 | 49.08 | 51.66 | 28.73 | 34.80 | 54.62 | 38.25 |
| 22k | 32.90 | 43.26 | 38.48 | 51.01 | 52.21 | 27.04 | 34.80 | 56.04 | 38.08 |
| 24k | 35.83 | 44.42 | 36.69 | 57.95 | 52.90 | 29.19 | 33.80 | 55.96 | 40.12 |
| 26k | 33.94 | 43.74 | 39.68 | 47.25 | 53.34 | 28.73 | 35.20 | 58.25 | 38.84 |
| 28k | 36.26 | 43.70 | 41.04 | 45.50 | 53.73 | 29.49 | 35.00 | 57.46 | 39.98 |
| 30k | 35.98 | 44.26 | 40.70 | 47.80 | 54.99 | 28.42 | 35.00 | 58.64 | 40.12 |
| 32k | 35.94 | 43.73 | 40.61 | 44.13 | 54.87 | 31.18 | 33.80 | 57.77 | 39.83 |
| 34k | 36.88 | 45.43 | 43.09 | 52.69 | 55.18 | 29.34 | 34.80 | 57.46 | 41.15 |
| 36k | 37.73 | 46.88 | 42.32 | 63.15 | 55.44 | 28.26 | 34.80 | 57.30 | 42.30 |
| 38k | 37.49 | 44.70 | 41.30 | 53.91 | 55.50 | 27.65 | 33.40 | 56.43 | 41.09 |
| 40k | 37.57 | 45.19 | 41.04 | 51.59 | 56.24 | 26.42 | 35.40 | 60.46 | 41.38 |
| 42k | 38.13 | 47.47 | 41.72 | 62.51 | 56.30 | 28.57 | 35.00 | 60.69 | 42.80 |
| 44k | 39.58 | 44.26 | 41.13 | 48.35 | 56.65 | 27.96 | 33.20 | 58.25 | 41.92 |
| 46k | 38.29 | 45.78 | 41.98 | 55.05 | 56.67 | 27.96 | 34.00 | 59.04 | 42.04 |
| 48k | 40.44 | 46.17 | 41.04 | 55.60 | 56.47 | 29.19 | 34.80 | 59.91 | 43.30 |
| 50k | 40.38 | 45.11 | 41.72 | 49.20 | 57.23 | 27.65 | 35.20 | 59.67 | 42.75 |
| 52k | 40.01 | 46.52 | 42.58 | 54.28 | 57.48 | 29.49 | 35.80 | 59.51 | 43.27 |
| 54k | 40.23 | 45.49 | 41.47 | 50.40 | 57.06 | 28.88 | 35.20 | 59.91 | 42.86 |
| 56k | 41.21 | 44.79 | 42.06 | 44.10 | 58.06 | 29.03 | 34.80 | 60.69 | 43.00 |
| 58k | 40.17 | 46.72 | 42.66 | 56.02 | 57.40 | 28.42 | 34.80 | 61.01 | 43.44 |
| 60k | 41.31 | 47.57 | 40.96 | 59.11 | 57.63 | 29.34 | 37.60 | 60.77 | 44.44 |
| 62k | 40.45 | 46.34 | 42.83 | 54.80 | 57.84 | 28.42 | 35.20 | 58.96 | 43.40 |
| 64k | 41.85 | 47.69 | 41.89 | 59.27 | 58.36 | 28.73 | 34.80 | 63.06 | 44.77 |
| 66k | 42.28 | 47.72 | 42.41 | 58.38 | 58.11 | 29.19 | 36.40 | 61.80 | 45.00 |
| 68k | 41.08 | 47.23 | 40.87 | 60.15 | 58.15 | 28.73 | 35.20 | 60.30 | 44.16 |
| 70k | 43.06 | 48.20 | 42.06 | 62.81 | 58.75 | 29.95 | 35.00 | 60.62 | 45.63 |
| 72k | 42.06 | 47.81 | 42.66 | 61.01 | 58.33 | 28.88 | 34.80 | 61.17 | 44.93 |
| 74k | 40.81 | 47.76 | 43.86 | 59.14 | 58.92 | 30.26 | 34.60 | 59.75 | 44.28 |
| 76k | 43.17 | 47.92 | 42.58 | 61.28 | 59.63 | 28.57 | 33.40 | 62.04 | 45.54 |
| 78k | 42.57 | 48.53 | 43.60 | 61.07 | 59.58 | 29.34 | 36.20 | 61.40 | 45.55 |
| 80k | 42.91 | 49.05 | 42.92 | 63.70 | 60.17 | 28.88 | 36.20 | 62.43 | 45.98 |
| 82k | 43.83 | 48.79 | 42.92 | 61.62 | 60.46 | 29.65 | 35.60 | 62.51 | 46.31 |
| 84k | 44.09 | 49.68 | 42.83 | 64.25 | 60.65 | 30.26 | 36.60 | 63.46 | 46.88 |
| 86k | 45.25 | 49.12 | 43.00 | 60.40 | 61.18 | 30.41 | 36.60 | 63.14 | 47.19 |
| 88k | 45.34 | 50.47 | 44.20 | 65.17 | 62.04 | 30.88 | 37.40 | 63.14 | 47.91 |
| 90k | 45.93 | 50.46 | 44.62 | 65.50 | 62.14 | 29.65 | 38.20 | 62.67 | 48.20 |
| 92k | 45.78 | 50.18 | 43.94 | 64.07 | 62.23 | 29.95 | 37.60 | 63.30 | 47.98 |
| 94k | 45.93 | 50.01 | 43.34 | 64.07 | 62.33 | 28.88 | 38.00 | 63.46 | 47.97 |
| 96k | 46.05 | 50.07 | 43.94 | 62.54 | 62.66 | 30.26 | 37.80 | 63.22 | 48.06 |
| 98k | 45.30 | 50.12 | 44.20 | 65.32 | 62.58 | 28.88 | 36.60 | 63.14 | 47.71 |

Table 4: Full evaluation results of 6B model pretraining in fig. 7, fig. 9 and fig. 10

| Steps | MMLU | QA_average | Arc_C | BoolQ | Hellaswag | Logiqa | Openbookqa | Winogrande | **Average** |
|---|---|---|---|---|---|---|---|---|---|
| 2k | 24.65 | 36.53 | 23.55 | 60.06 | 28.27 | 27.50 | 28.00 | 51.78 | 30.59 |
| 4k | 27.03 | 38.87 | 29.95 | 59.79 | 37.24 | 27.96 | 27.60 | 50.67 | 32.95 |
| 6k | 30.12 | 39.68 | 32.85 | 50.24 | 42.74 | 27.65 | 32.80 | 51.78 | 34.90 |
| 8k | 31.76 | 42.20 | 36.52 | 52.35 | 47.67 | 29.49 | 32.40 | 54.78 | 36.98 |
| 10k | 30.73 | 44.68 | 38.65 | 61.47 | 50.44 | 28.73 | 32.60 | 56.20 | 37.71 |
| 12k | 33.63 | 44.83 | 38.91 | 59.08 | 51.98 | 28.11 | 34.80 | 56.12 | 39.23 |
| 14k | 33.07 | 45.23 | 41.30 | 56.73 | 53.21 | 27.96 | 34.80 | 57.38 | 39.15 |
| 16k | 32.82 | 44.77 | 40.78 | 51.62 | 55.11 | 26.42 | 35.80 | 58.88 | 38.79 |
| 18k | 36.58 | 46.53 | 40.96 | 59.42 | 55.21 | 28.88 | 36.40 | 58.33 | 41.56 |
| 20k | 35.78 | 43.18 | 41.30 | 40.86 | 56.51 | 27.80 | 35.80 | 56.83 | 39.48 |
| 22k | 38.06 | 45.40 | 42.24 | 48.47 | 57.30 | 29.34 | 35.20 | 59.83 | 41.73 |
| 24k | 39.20 | 46.05 | 42.32 | 54.19 | 57.59 | 26.42 | 37.00 | 58.80 | 42.63 |
| 26k | 38.11 | 45.80 | 42.41 | 49.91 | 58.19 | 28.42 | 35.80 | 60.06 | 41.95 |
| 28k | 40.93 | 45.01 | 40.53 | 46.27 | 59.00 | 28.42 | 36.40 | 59.43 | 42.97 |
| 30k | 42.24 | 45.79 | 43.60 | 43.21 | 59.41 | 28.42 | 38.20 | 61.88 | 44.01 |
| 32k | 42.28 | 47.92 | 44.37 | 56.39 | 59.59 | 29.49 | 37.20 | 60.46 | 45.10 |
| 34k | 42.10 | 46.44 | 44.20 | 48.84 | 59.96 | 28.42 | 38.60 | 58.64 | 44.27 |
| 36k | 43.63 | 46.43 | 44.88 | 44.74 | 60.47 | 29.34 | 38.00 | 61.17 | 45.03 |
| 38k | 42.81 | 47.64 | 45.73 | 48.41 | 61.11 | 29.03 | 38.00 | 63.54 | 45.22 |
| 40k | 44.02 | 47.77 | 45.65 | 51.56 | 60.83 | 28.88 | 39.40 | 60.30 | 45.90 |
| 42k | 44.30 | 49.44 | 45.56 | 62.35 | 61.41 | 29.03 | 37.80 | 60.46 | 46.87 |
| 44k | 44.59 | 47.44 | 45.31 | 48.41 | 61.66 | 29.19 | 38.40 | 61.64 | 46.01 |
| 46k | 44.51 | 47.27 | 46.50 | 46.51 | 61.81 | 28.26 | 37.40 | 63.14 | 45.89 |

Table 5: Full evaluation results of 6B model interpositional growth at 24k in fig. 3, fig. 9 and fig. 10

| Steps | MMLU | QA_average | Arc_C | BoolQ | Hellaswag | Logiqa | Openbookqa | Winogrande | **Average** |
|---|---|---|---|---|---|---|---|---|---|
| 24k | 34.22 | 44.57 | 37.80 | 62.45 | 50.90 | 27.80 | 33.20 | 55.25 | 39.39 |
| 26k | 35.86 | 45.03 | 39.76 | 52.59 | 54.62 | 30.57 | 35.00 | 57.62 | 40.44 |
| 28k | 36.53 | 45.21 | 40.53 | 52.02 | 55.76 | 29.19 | 34.80 | 58.96 | 40.87 |
| 30k | 39.30 | 45.40 | 41.04 | 51.62 | 56.94 | 28.42 | 35.60 | 58.80 | 42.35 |
| 32k | 40.50 | 45.73 | 42.24 | 51.62 | 57.67 | 29.19 | 35.20 | 58.48 | 43.12 |
| 34k | 41.50 | 46.61 | 43.34 | 53.76 | 57.56 | 29.34 | 35.80 | 59.83 | 44.05 |
| 36k | 42.38 | 48.17 | 42.24 | 62.11 | 58.75 | 27.96 | 36.40 | 61.56 | 45.28 |
| 38k | 43.46 | 48.53 | 44.28 | 61.56 | 58.85 | 28.73 | 37.00 | 60.77 | 46.00 |
| 40k | 42.61 | 48.75 | 43.17 | 63.79 | 59.31 | 28.73 | 35.80 | 61.72 | 45.68 |
| 42k | 44.30 | 49.22 | 45.22 | 65.60 | 59.79 | 28.11 | 36.00 | 60.62 | 46.76 |
| 44k | 44.30 | 48.73 | 46.08 | 58.93 | 60.06 | 30.57 | 37.20 | 59.51 | 46.51 |
| 46k | 45.11 | 48.48 | 44.28 | 56.24 | 60.71 | 30.72 | 36.20 | 62.75 | 46.80 |
| 48k | 46.03 | 49.36 | 44.88 | 62.66 | 60.33 | 28.42 | 36.80 | 63.06 | 47.69 |
| 50k | 44.54 | 49.38 | 43.43 | 61.87 | 61.11 | 30.88 | 37.20 | 61.80 | 46.96 |
| 52k | 46.52 | 49.47 | 45.05 | 57.92 | 61.23 | 30.88 | 38.20 | 63.54 | 48.00 |
| 54k | 46.99 | 48.61 | 46.16 | 51.10 | 62.19 | 30.26 | 39.20 | 62.75 | 47.80 |

Table 6: Full evaluation results of 6B model stack growth at 24k in fig. 3

| Steps | MMLU | QA_average | Arc_C | BoolQ | Hellaswag | Logiqa | Openbookqa | Winogrande | **Average** |
|---|---|---|---|---|---|---|---|---|---|
| 24k | 34.14 | 44.94 | 37.20 | 60.24 | 52.35 | 28.57 | 35.40 | 55.88 | 39.54 |
| 26k | 33.69 | 44.83 | 41.21 | 50.73 | 54.61 | 29.95 | 35.00 | 57.46 | 39.26 |
| 28k | 34.77 | 45.31 | 41.38 | 54.65 | 55.60 | 28.88 | 33.40 | 57.93 | 40.04 |
| 30k | 36.41 | 46.14 | 41.30 | 57.55 | 56.67 | 29.03 | 35.00 | 57.30 | 41.28 |
| 32k | 37.54 | 44.92 | 42.92 | 49.17 | 56.88 | 28.57 | 34.20 | 57.77 | 41.23 |
| 34k | 36.68 | 46.71 | 43.09 | 54.62 | 57.86 | 28.57 | 36.20 | 59.91 | 41.69 |
| 36k | 39.20 | 47.42 | 43.26 | 58.13 | 58.34 | 29.19 | 36.40 | 59.19 | 43.31 |
| 38k | 39.70 | 47.61 | 43.34 | 60.55 | 58.92 | 29.65 | 34.00 | 59.19 | 43.65 |
| 40k | 36.92 | 47.39 | 42.75 | 55.96 | 59.40 | 30.72 | 36.00 | 59.51 | 42.16 |

Table 7: Full evaluation results of 6B model interpositional growth at 8k in fig. 9 and fig. 10

| Steps | MMLU | QA_average | Arc_C | BoolQ | Hellaswag | Logiqa | Openbookqa | Winogrande | **Average** |
|---|---|---|---|---|---|---|---|---|---|
| 8k | 29.75 | 41.58 | 34.30 | 59.72 | 43.29 | 29.65 | 29.80 | 52.72 | 35.67 |
| 10k | 30.40 | 41.91 | 34.98 | 52.48 | 46.70 | 29.19 | 33.20 | 54.93 | 36.16 |
| 12k | 32.64 | 41.68 | 35.67 | 48.69 | 49.62 | 27.34 | 32.40 | 56.35 | 37.16 |
| 14k | 33.35 | 44.27 | 37.29 | 56.45 | 51.87 | 27.34 | 34.60 | 58.09 | 38.81 |
| 16k | 33.95 | 45.13 | 40.61 | 54.10 | 53.55 | 30.57 | 35.60 | 56.35 | 39.54 |
| 18k | 36.41 | 46.74 | 40.02 | 61.80 | 54.59 | 29.95 | 36.00 | 58.09 | 41.58 |
| 20k | 38.53 | 44.96 | 40.02 | 54.07 | 55.06 | 27.65 | 34.80 | 58.17 | 41.75 |
| 22k | 38.33 | 45.80 | 42.66 | 54.10 | 56.27 | 28.26 | 35.60 | 57.93 | 42.07 |
| 24k | 38.72 | 47.38 | 42.92 | 59.57 | 57.15 | 29.34 | 35.00 | 60.30 | 43.05 |
| 26k | 39.78 | 48.56 | 42.92 | 64.19 | 57.44 | 29.49 | 36.40 | 60.93 | 44.17 |
| 28k | 41.53 | 46.35 | 43.86 | 50.95 | 58.15 | 28.73 | 36.00 | 60.38 | 43.94 |
| 30k | 41.40 | 46.74 | 42.06 | 52.72 | 58.75 | 29.19 | 35.60 | 62.12 | 44.07 |
| 32k | 43.13 | 48.44 | 42.83 | 61.16 | 58.79 | 29.65 | 36.80 | 61.40 | 45.78 |
| 34k | 42.12 | 46.95 | 42.49 | 51.22 | 59.62 | 29.80 | 35.40 | 63.14 | 44.53 |
| 36k | 44.05 | 48.12 | 43.60 | 58.69 | 60.23 | 30.57 | 34.80 | 60.85 | 46.09 |
| 38k | 44.70 | 48.44 | 44.03 | 58.47 | 60.69 | 27.96 | 37.20 | 62.27 | 46.57 |
| 40k | 43.66 | 48.69 | 46.08 | 56.97 | 60.59 | 29.65 | 36.80 | 62.04 | 46.17 |
| 42k | 44.75 | 49.35 | 45.48 | 59.76 | 60.59 | 29.65 | 38.00 | 62.59 | 47.05 |
| 44k | 45.46 | 49.23 | 44.71 | 60.76 | 60.88 | 29.95 | 36.40 | 62.67 | 47.34 |
| 46k | 45.39 | 48.94 | 45.82 | 57.46 | 61.25 | 29.49 | 36.80 | 62.80 | 47.16 |
| 48k | 46.15 | 49.18 | 44.45 | 57.37 | 61.51 | 30.88 | 37.80 | 63.06 | 47.66 |

Table 8: Full evaluation results of 6B model interpositional growth at 16k in fig. 9 and fig. 10

| Steps | MMLU | QA_average | Arc_C | BoolQ | Hellaswag | Logiqa | Openbookqa | Winogrande | **Average** |
|---|---|---|---|---|---|---|---|---|---|
| 16k | 31.23 | 44.10 | 36.35 | 60.73 | 48.80 | 30.72 | 32.80 | 55.17 | 37.66 |
| 18k | 34.09 | 43.29 | 37.12 | 50.76 | 51.92 | 30.41 | 33.40 | 56.12 | 38.69 |
| 20k | 33.76 | 44.06 | 38.74 | 50.06 | 53.70 | 29.19 | 34.80 | 57.85 | 38.91 |
| 22k | 37.69 | 46.01 | 40.96 | 54.13 | 55.20 | 31.18 | 35.40 | 59.19 | 41.85 |
| 24k | 38.04 | 44.89 | 40.61 | 47.52 | 55.93 | 30.57 | 36.20 | 58.48 | 41.46 |
| 26k | 39.81 | 44.63 | 41.30 | 47.80 | 56.48 | 27.80 | 35.20 | 59.19 | 42.22 |
| 28k | 41.23 | 46.99 | 40.78 | 60.37 | 57.35 | 28.73 | 36.20 | 58.48 | 44.11 |
| 30k | 41.65 | 47.65 | 43.34 | 57.92 | 57.92 | 29.49 | 36.40 | 60.85 | 44.65 |
| 32k | 41.41 | 48.63 | 43.00 | 62.51 | 58.41 | 29.95 | 37.60 | 60.30 | 45.02 |
| 34k | 42.89 | 48.06 | 43.69 | 59.88 | 58.63 | 29.65 | 38.20 | 58.33 | 45.48 |
| 36k | 43.57 | 47.32 | 43.77 | 56.12 | 58.66 | 28.11 | 37.80 | 59.43 | 45.44 |
| 38k | 44.87 | 49.06 | 43.09 | 62.54 | 59.41 | 30.26 | 36.80 | 62.27 | 46.97 |
| 40k | 45.24 | 49.02 | 43.52 | 63.64 | 59.86 | 27.34 | 36.80 | 62.98 | 47.13 |
| 42k | 44.01 | 48.44 | 43.34 | 57.19 | 60.52 | 29.49 | 38.00 | 62.12 | 46.23 |
| 44k | 45.34 | 46.91 | 42.92 | 49.63 | 60.62 | 29.49 | 37.00 | 61.80 | 46.13 |
| 46k | 45.22 | 49.51 | 45.39 | 57.68 | 61.23 | 32.41 | 38.40 | 61.96 | 47.37 |
| 48k | 45.70 | 49.53 | 44.97 | 60.43 | 61.34 | 28.88 | 37.60 | 63.93 | 47.61 |
| 50k | 46.21 | 50.64 | 45.73 | 63.27 | 61.37 | 30.88 | 38.00 | 64.56 | 48.42 |
| 52k | 46.05 | 51.01 | 46.25 | 67.65 | 61.73 | 30.41 | 37.80 | 62.19 | 48.53 |

Table 9: Full evaluation results of 6B model interpositional growth at 32k in fig. 9 and fig. 10

| Steps | MMLU | QA_average | Arc_C | BoolQ | Hellaswag | Logiqa | Openbookqa | Winogrande | **Average** |
|---|---|---|---|---|---|---|---|---|---|
| 32k | 34.45 | 43.64 | 40.61 | 49.02 | 53.48 | 28.88 | 33.20 | 56.67 | 39.05 |
| 34k | 36.84 | 44.41 | 40.96 | 50.80 | 56.41 | 27.65 | 33.60 | 57.06 | 40.63 |
| 36k | 38.76 | 46.45 | 41.13 | 54.46 | 57.11 | 30.57 | 35.20 | 60.22 | 42.60 |
| 38k | 39.95 | 47.68 | 43.26 | 61.44 | 58.15 | 30.11 | 34.80 | 58.33 | 43.82 |
| 40k | 41.29 | 46.62 | 42.49 | 55.05 | 58.30 | 28.88 | 35.00 | 59.98 | 43.95 |
| 42k | 41.21 | 47.39 | 40.87 | 61.31 | 58.95 | 28.42 | 35.80 | 58.96 | 44.30 |
| 44k | 43.70 | 47.55 | 43.17 | 57.03 | 59.91 | 29.19 | 36.00 | 60.01 | 45.63 |
| 46k | 44.30 | 49.20 | 45.05 | 64.65 | 59.75 | 29.03 | 36.20 | 60.54 | 46.75 |
| 48k | 43.01 | 48.67 | 44.28 | 61.65 | 60.09 | 27.96 | 36.80 | 61.25 | 45.84 |
| 50k | 45.24 | 48.64 | 44.37 | 62.23 | 60.39 | 29.49 | 35.00 | 60.38 | 46.94 |
| 52k | 45.68 | 48.67 | 45.82 | 56.61 | 60.52 | 30.72 | 37.60 | 60.77 | 47.18 |
| 54k | 45.15 | 49.75 | 44.03 | 65.32 | 61.01 | 28.73 | 36.40 | 62.98 | 47.45 |
| 56k | 46.18 | 50.12 | 44.80 | 67.71 | 60.88 | 26.88 | 37.40 | 63.06 | 48.15 |

Table 10: Full evaluation results of 6B model interpositional growth at 40k in fig. 9 and fig. 10

| Steps | MMLU | QA_average | Arc_C | BoolQ | Hellaswag | Logiqa | Openbookqa | Winogrande | **Average** |
|---|---|---|---|---|---|---|---|---|---|
| 40k | 36.87 | 45.36 | 39.59 | 59.08 | 54.18 | 26.88 | 34.00 | 58.41 | 41.11 |
| 42k | 38.41 | 46.43 | 41.72 | 53.94 | 57.27 | 30.57 | 35.80 | 59.27 | 42.42 |
| 44k | 40.68 | 46.91 | 42.06 | 56.70 | 58.33 | 28.73 | 35.20 | 60.46 | 43.80 |
| 46k | 41.70 | 47.79 | 43.00 | 62.26 | 58.92 | 28.42 | 34.40 | 59.75 | 44.75 |
| 48k | 43.22 | 47.28 | 43.77 | 54.50 | 59.24 | 26.88 | 36.40 | 62.90 | 45.25 |
| 50k | 42.74 | 47.91 | 43.26 | 56.21 | 59.66 | 30.11 | 36.40 | 61.80 | 45.32 |
| 52k | 44.47 | 48.39 | 43.60 | 57.28 | 60.60 | 29.80 | 36.60 | 62.43 | 46.43 |
| 54k | 45.73 | 48.27 | 43.17 | 61.53 | 60.49 | 28.57 | 34.00 | 61.88 | 47.00 |
| 56k | 44.91 | 49.11 | 43.86 | 63.46 | 60.88 | 28.57 | 35.40 | 62.51 | 47.01 |
| 58k | 46.55 | 50.09 | 45.31 | 63.61 | 61.25 | 29.03 | 38.80 | 62.51 | 48.32 |
| 60k | 46.34 | 49.05 | 44.88 | 58.72 | 61.50 | 31.64 | 36.40 | 61.17 | 47.70 |

Table 11: Full evaluation results of 6B model interpositional growth at 48k in fig. 9 and fig. 10

| Steps | MMLU | QA_average | Arc_C | BoolQ | Hellaswag | Logiqa | Openbookqa | Winogrande | **Average** |
|---|---|---|---|---|---|---|---|---|---|
| 48k | 38.14 | 46.38 | 41.81 | 61.01 | 54.74 | 27.80 | 35.80 | 57.14 | 42.26 |
| 50k | 40.88 | 47.66 | 42.56 | 61.56 | 57.46 | 30.15 | 34.61 | 59.64 | 44.27 |
| 52k | 41.65 | 48.64 | 43.00 | 63.85 | 58.71 | 31.03 | 34.40 | 60.85 | 45.15 |
| 54k | 43.24 | 48.76 | 43.86 | 61.47 | 59.53 | 30.11 | 36.60 | 61.01 | 46.00 |
| 56k | 44.23 | 49.00 | 43.77 | 63.12 | 59.71 | 28.88 | 36.40 | 62.12 | 46.62 |
| 58k | 43.23 | 48.87 | 43.60 | 59.36 | 60.35 | 29.95 | 36.60 | 63.38 | 46.05 |
| 60k | 44.95 | 50.67 | 45.73 | 65.57 | 60.89 | 30.72 | 38.00 | 63.12 | 47.81 |
| 62k | 44.98 | 49.41 | 44.20 | 64.43 | 61.23 | 28.26 | 36.40 | 61.96 | 47.20 |

Table 12: Full evaluation results of 6B model interpositional growth at 56k in fig. 9 and fig. 10

| Steps | MMLU | QA_average | Arc_C | BoolQ | Hellaswag | Logiqa | Openbookqa | Winogrande | **Average** |
|---|---|---|---|---|---|---|---|---|---|
| 56k | 40.41 | 47.31 | 41.47 | 63.49 | 55.81 | 28.11 | 35.40 | 59.59 | 43.86 |
| 58k | 42.44 | 46.71 | 43.60 | 54.16 | 58.95 | 29.65 | 35.00 | 58.88 | 44.57 |
| 60k | 43.17 | 48.28 | 44.37 | 61.28 | 59.30 | 28.73 | 34.80 | 61.17 | 45.72 |
| 62k | 42.81 | 49.61 | 44.28 | 65.75 | 60.23 | 29.03 | 36.80 | 61.56 | 46.21 |
| 64k | 45.24 | 48.86 | 44.03 | 59.54 | 60.37 | 31.03 | 36.40 | 61.80 | 47.05 |
| 66k | 44.77 | 49.29 | 46.76 | 58.62 | 61.14 | 29.95 | 37.60 | 61.64 | 47.03 |
| 68k | 45.11 | 49.68 | 42.75 | 66.15 | 61.41 | 27.80 | 36.80 | 63.14 | 47.39 |
| 70k | 45.36 | 49.38 | 42.68 | 65.38 | 60.45 | 29.70 | 36.20 | 61.89 | 47.37 |

Table 13: Full evaluation results of 6B model interpositional growth at 64k in fig. 9 and fig. 10

| Steps | MMLU | QA_average | Arc_C | BoolQ | Hellaswag | Logiqa | Openbookqa | Winogrande | **Average** |
|---|---|---|---|---|---|---|---|---|---|
| 64k | 39.47 | 47.17 | 42.41 | 59.97 | 55.83 | 29.19 | 35.00 | 60.62 | 43.32 |
| 66k | 43.46 | 47.74 | 42.92 | 57.03 | 59.50 | 29.19 | 36.60 | 61.17 | 45.60 |
| 68k | 43.93 | 48.95 | 44.80 | 61.93 | 59.61 | 29.34 | 35.80 | 62.19 | 46.44 |
| 70k | 43.36 | 48.89 | 44.03 | 62.26 | 60.37 | 28.88 | 34.40 | 63.38 | 46.12 |
| 72k | 45.09 | 49.44 | 45.48 | 62.29 | 60.43 | 29.80 | 35.60 | 63.06 | 47.27 |
| 74k | 44.92 | 49.44 | 45.05 | 58.72 | 61.35 | 31.34 | 37.60 | 62.59 | 47.18 |
| 76k | 46.00 | 50.34 | 44.80 | 67.06 | 61.67 | 29.03 | 36.80 | 62.67 | 48.17 |
| 78k | 45.67 | 49.95 | 43.68 | 66.48 | 60.59 | 30.50 | 36.70 | 61.74 | 47.81 |

Table 14: Full evaluation results of 6B model interpositional growth at 72k in fig. 9

| Steps | MMLU | QA_average | Arc_C | BoolQ | Hellaswag | Logiqa | Openbookqa | Winogrande | **Average** |
|---|---|---|---|---|---|---|---|---|---|
| 72k | 40.44 | 46.88 | 42.24 | 60.52 | 55.79 | 29.03 | 34.40 | 59.27 | 43.66 |
| 74k | 44.35 | 49.47 | 45.56 | 65.17 | 59.75 | 29.95 | 35.40 | 61.01 | 46.91 |
| 76k | 44.97 | 49.50 | 45.31 | 67.28 | 60.02 | 30.11 | 34.00 | 60.30 | 47.24 |
| 78k | 44.45 | 48.94 | 44.45 | 61.68 | 60.89 | 28.57 | 34.80 | 63.22 | 46.69 |
| 80k | 45.33 | 50.27 | 46.08 | 65.05 | 60.88 | 29.80 | 36.20 | 63.61 | 47.80 |
| 82k | 45.27 | 50.97 | 45.05 | 66.79 | 61.67 | 31.34 | 37.60 | 63.38 | 48.12 |
| 84k | 46.61 | 50.76 | 45.14 | 67.92 | 61.83 | 29.49 | 35.80 | 64.40 | 48.69 |
| 86k | 46.32 | 49.48 | 45.14 | 61.16 | 61.91 | 29.03 | 35.40 | 64.25 | 47.90 |

Table 15: Full evaluation results of 6B model interpositional growth at 80k in fig. 9

| Steps | MMLU | QA_average | Arc_C | BoolQ | Hellaswag | Logiqa | Openbookqa | Winogrande | **Average** |
|---|---|---|---|---|---|---|---|---|---|
| 80k | 41.46 | 48.44 | 41.64 | 65.54 | 57.39 | 28.73 | 36.00 | 61.33 | 44.95 |
| 82k | 44.68 | 49.58 | 44.03 | 62.11 | 60.33 | 30.57 | 37.60 | 62.83 | 47.13 |
| 84k | 45.46 | 49.17 | 44.28 | 63.98 | 60.76 | 29.34 | 36.20 | 60.46 | 47.32 |
| 86k | 43.46 | 49.56 | 45.65 | 64.53 | 61.17 | 28.26 | 35.80 | 61.96 | 46.51 |
| 88k | 46.30 | 49.27 | 44.97 | 61.13 | 61.03 | 27.50 | 36.40 | 64.56 | 47.78 |
| 90k | 45.86 | 50.71 | 44.88 | 67.09 | 61.78 | 29.95 | 37.20 | 63.38 | 48.29 |
| 92k | 46.86 | 50.46 | 44.03 | 65.47 | 62.63 | 29.49 | 37.60 | 63.54 | 48.66 |
| 94k | 47.12 | 50.39 | 45.39 | 65.57 | 62.43 | 28.57 | 36.00 | 64.40 | 48.76 |

Table 16: Full evaluation results of 6B model interpositional growth at 88k in fig. 9

| Steps | MMLU | QA_average | Arc_C | BoolQ | Hellaswag | Logiqa | Openbookqa | Winogrande | **Average** |
|---|---|---|---|---|---|---|---|---|---|
| 88k | 43.78 | 49.09 | 44.45 | 63.85 | 58.70 | 29.49 | 35.60 | 62.43 | 46.43 |
| 90k | 44.52 | 49.75 | 43.26 | 64.01 | 60.38 | 30.11 | 37.60 | 63.14 | 47.14 |
| 92k | 45.76 | 48.43 | 45.22 | 58.90 | 60.69 | 29.95 | 35.20 | 60.62 | 47.10 |
| 94k | 43.42 | 50.46 | 45.22 | 67.82 | 61.11 | 29.19 | 35.80 | 63.61 | 46.94 |
| 96k | 45.95 | 50.73 | 44.88 | 68.59 | 61.10 | 29.80 | 37.20 | 62.83 | 48.34 |
| 98k | 45.59 | 51.92 | 46.33 | 68.81 | 62.13 | 31.95 | 38.60 | 63.69 | 48.75 |
| 100k | 47.80 | 50.93 | 44.20 | 66.09 | 62.52 | 30.88 | 36.80 | 65.11 | 49.37 |
| 102k | 47.52 | 50.45 | 45.05 | 66.91 | 62.44 | 27.96 | 37.20 | 63.14 | 48.99 |

Table 17: Full evaluation results of 6B model interpositional growth at 96k in fig. 9

| Steps | MMLU | QA_average | Arc_C | BoolQ | Hellaswag | Logiqa | Openbookqa | Winogrande | **Average** |
|---|---|---|---|---|---|---|---|---|---|
| 96k | 43.80 | 48.58 | 43.52 | 63.58 | 58.94 | 28.57 | 35.60 | 61.25 | 46.19 |
| 98k | 45.05 | 49.03 | 43.77 | 57.49 | 60.76 | 30.72 | 37.60 | 63.85 | 47.04 |
| 100k | 45.28 | 49.44 | 44.54 | 64.59 | 60.84 | 30.57 | 35.00 | 61.09 | 47.36 |
| 102k | 44.49 | 50.30 | 45.56 | 66.57 | 60.69 | 29.95 | 36.60 | 62.43 | 47.40 |
| 104k | 45.94 | 50.56 | 45.56 | 64.68 | 61.45 | 29.65 | 37.60 | 64.40 | 48.25 |
| 106k | 45.88 | 51.12 | 45.56 | 65.93 | 61.55 | 30.57 | 37.60 | 65.51 | 48.50 |
| 108k | 47.41 | 51.11 | 44.80 | 67.89 | 62.31 | 29.19 | 38.00 | 64.48 | 49.26 |
| 110k | 46.96 | 50.08 | 44.71 | 65.66 | 62.25 | 28.57 | 36.00 | 63.30 | 48.52 |

Table 18: Full evaluation results of 6B model width growth with no noise in fig. 6

| Steps | MMLU | QA_average | Arc_C | BoolQ | Hellaswag | Logiqa | Openbookqa | Winogrande | **Average** |
|---|---|---|---|---|---|---|---|---|---|
| 24k | 35.85 | 44.65 | 36.86 | 58.10 | 52.79 | 29.34 | 33.80 | 56.99 | 40.25 |
| 26k | 33.89 | 44.02 | 39.51 | 48.10 | 54.52 | 29.49 | 35.60 | 56.87 | 38.95 |
| 28k | 35.22 | 43.14 | 40.61 | 45.78 | 54.67 | 28.26 | 33.40 | 56.12 | 39.18 |
| 30k | 36.35 | 45.00 | 41.04 | 52.39 | 55.99 | 30.57 | 33.40 | 56.59 | 40.67 |
| 32k | 37.60 | 45.61 | 42.49 | 55.02 | 56.11 | 28.26 | 35.60 | 56.20 | 41.61 |
| 34k | 37.56 | 45.98 | 42.41 | 55.26 | 56.61 | 28.42 | 36.80 | 56.35 | 41.77 |
| 36k | 38.38 | 44.37 | 40.78 | 43.84 | 58.02 | 27.80 | 36.40 | 59.35 | 41.37 |
| 38k | 39.83 | 46.66 | 43.26 | 54.89 | 58.00 | 29.34 | 35.20 | 59.27 | 43.25 |
| 40k | 38.78 | 48.59 | 44.62 | 63.12 | 58.79 | 27.96 | 36.40 | 60.62 | 43.68 |
| 42k | 41.30 | 47.69 | 45.48 | 57.40 | 59.16 | 27.96 | 36.60 | 59.51 | 44.49 |
| 44k | 41.35 | 47.65 | 44.88 | 56.73 | 59.37 | 29.80 | 36.40 | 58.72 | 44.50 |

Table 19: Full evaluation results of 6B model width growth with noise std=0.01 in fig. 6 and fig. 7

| Steps | MMLU | QA_average | Arc_C | BoolQ | Hellaswag | Logiqa | Openbookqa | Winogrande | **Average** |
|---|---|---|---|---|---|---|---|---|---|
| 24k | 35.91 | 44.52 | 36.86 | 58.23 | 52.79 | 28.42 | 33.80 | 56.99 | 40.21 |
| 26k | 33.43 | 43.80 | 39.59 | 47.25 | 54.30 | 28.88 | 35.40 | 57.38 | 38.62 |
| 28k | 33.58 | 44.12 | 40.96 | 48.38 | 55.36 | 29.03 | 33.20 | 57.77 | 38.85 |
| 30k | 36.55 | 46.73 | 41.47 | 61.77 | 56.13 | 30.41 | 33.60 | 56.99 | 41.64 |
| 32k | 37.64 | 46.76 | 42.92 | 57.95 | 56.81 | 28.57 | 35.40 | 58.88 | 42.20 |
| 34k | 38.33 | 46.72 | 42.75 | 57.86 | 57.39 | 28.42 | 35.80 | 58.09 | 42.52 |
| 36k | 39.99 | 45.28 | 39.33 | 51.31 | 58.12 | 29.19 | 36.40 | 57.30 | 42.63 |
| 38k | 41.46 | 48.20 | 43.52 | 63.09 | 58.48 | 27.50 | 38.20 | 58.41 | 44.83 |
| 40k | 39.11 | 47.81 | 43.52 | 59.85 | 58.64 | 29.34 | 36.60 | 58.88 | 43.46 |
| 42k | 41.36 | 48.27 | 43.86 | 61.74 | 59.24 | 28.57 | 37.00 | 59.19 | 44.81 |
| 44k | 42.47 | 48.56 | 44.28 | 61.53 | 59.56 | 29.34 | 36.20 | 60.46 | 45.52 |

Table 20: Full evaluation results of 6B model width growth with noise std=0.05 in fig. 6

| Steps | MMLU | QA_average | Arc_C | BoolQ | Hellaswag | Logiqa | Openbookqa | Winogrande | **Average** |
|---|---|---|---|---|---|---|---|---|---|
| 24k | 36.03 | 44.81 | 37.03 | 58.47 | 52.72 | 29.34 | 34.00 | 57.30 | 40.42 |
| 26k | 33.94 | 43.63 | 39.08 | 46.94 | 54.30 | 29.95 | 34.20 | 57.30 | 38.78 |
| 28k | 35.60 | 45.12 | 41.72 | 55.57 | 54.94 | 29.03 | 32.80 | 56.67 | 40.36 |
| 30k | 36.87 | 46.48 | 41.72 | 61.35 | 56.09 | 29.65 | 34.20 | 55.88 | 41.68 |
| 32k | 37.89 | 46.45 | 42.75 | 57.37 | 56.84 | 29.49 | 35.00 | 57.22 | 42.17 |
| 34k | 38.30 | 47.89 | 42.41 | 62.54 | 57.00 | 29.03 | 38.60 | 57.77 | 43.10 |
| 36k | 39.65 | 46.35 | 40.27 | 56.57 | 58.26 | 28.42 | 36.00 | 58.56 | 43.00 |
| 38k | 40.02 | 47.57 | 43.00 | 59.85 | 58.62 | 30.41 | 35.00 | 58.56 | 43.80 |
| 40k | 39.72 | 47.93 | 43.43 | 61.10 | 59.00 | 26.88 | 37.60 | 59.59 | 43.83 |
| 42k | 41.85 | 48.21 | 44.28 | 60.34 | 59.16 | 30.11 | 36.80 | 58.56 | 45.03 |
| 44k | 42.41 | 48.42 | 45.31 | 60.06 | 59.39 | 29.65 | 36.20 | 59.91 | 45.42 |

Table 21: Full evaluation results of 6B model width growth with noise std=0.1 in fig. 6

| Steps | MMLU | QA_average | Arc_C | BoolQ | Hellaswag | Logiqa | Openbookqa | Winogrande | **Average** |
|---|---|---|---|---|---|---|---|---|---|
| 24k | 35.84 | 44.67 | 36.86 | 58.72 | 52.80 | 28.88 | 34.40 | 56.35 | 40.25 |
| 26k | 34.38 | 43.77 | 38.99 | 46.97 | 54.33 | 29.19 | 34.80 | 58.33 | 39.07 |
| 28k | 35.90 | 44.10 | 41.98 | 47.03 | 55.28 | 29.19 | 33.60 | 57.54 | 40.00 |
| 30k | 37.10 | 46.17 | 41.30 | 60.24 | 56.30 | 29.80 | 33.00 | 56.35 | 41.63 |
| 32k | 38.24 | 47.00 | 43.94 | 61.56 | 56.39 | 28.42 | 34.40 | 57.30 | 42.62 |
| 34k | 38.26 | 47.23 | 43.26 | 61.93 | 57.14 | 28.11 | 36.00 | 56.91 | 42.74 |
| 36k | 40.09 | 45.85 | 42.06 | 54.16 | 58.16 | 26.73 | 36.00 | 58.01 | 42.97 |
| 38k | 41.33 | 46.48 | 43.17 | 54.62 | 58.43 | 29.95 | 35.00 | 57.70 | 43.90 |
| 40k | 39.58 | 48.60 | 44.28 | 62.87 | 59.07 | 28.88 | 37.00 | 59.51 | 44.09 |
| 42k | 41.98 | 48.11 | 43.77 | 61.38 | 59.02 | 28.88 | 36.20 | 59.43 | 45.05 |
| 44k | 41.69 | 48.18 | 44.71 | 57.16 | 59.42 | 29.65 | 37.20 | 60.93 | 44.93 |

Table 22: Full evaluation results of 6B models direct growth under different model structure in fig. 7

| Model | Steps | MMLU | QA_average | Arc_C | BoolQ | Hellaswag | Logiqa | Openbookqa | Winogrande | **Average** |
|---|---|---|---|---|---|---|---|---|---|---|
| 3B Pre-norm | 8k | 29.19 | 41.26 | 34.56 | 53.36 | 43.62 | 29.65 | 32.60 | 53.75 | 35.22 |
| | 16k | 32.07 | 44.10 | 37.03 | 56.79 | 50.15 | 29.65 | 34.40 | 56.59 | 38.09 |
| | 24k | 35.83 | 44.42 | 36.69 | 57.95 | 52.90 | 29.19 | 33.80 | 55.96 | 40.12 |
| | 32k | 36.21 | 43.73 | 40.61 | 44.13 | 54.87 | 31.18 | 33.80 | 57.77 | 39.97 |
| 6B Pre-norm Depth | 8k | 29.75 | 41.58 | 34.30 | 59.70 | 43.29 | 29.65 | 29.80 | 52.72 | 35.66 |
| | 16k | 31.23 | 44.10 | 36.35 | 60.73 | 48.80 | 30.72 | 32.80 | 55.17 | 37.66 |
| | 24k | 34.22 | 44.57 | 37.80 | 62.45 | 50.90 | 27.80 | 33.20 | 55.25 | 39.39 |
| | 32k | 34.45 | 43.64 | 40.61 | 49.02 | 53.48 | 28.88 | 33.20 | 56.67 | 39.05 |
| 6B Pre-norm Width | 8k | 30.71 | 41.53 | 34.81 | 54.04 | 43.58 | 30.26 | 32.40 | 54.06 | 36.12 |
| | 16k | 33.23 | 44.04 | 37.03 | 56.79 | 50.13 | 30.26 | 34.00 | 56.04 | 38.64 |
| | 24k | 35.99 | 44.68 | 37.29 | 58.07 | 52.73 | 29.34 | 33.80 | 56.83 | 40.33 |
| | 32k | 35.84 | 43.61 | 40.53 | 44.22 | 54.88 | 30.88 | 33.40 | 57.77 | 39.73 |
| 3B Post-norm | 8k | 28.85 | 40.06 | 29.69 | 59.48 | 41.59 | 26.88 | 30.00 | 52.72 | 34.46 |
| | 16k | 30.17 | 42.27 | 33.02 | 56.33 | 47.01 | 29.95 | 32.60 | 54.70 | 36.22 |
| | 24k | 32.64 | 42.18 | 33.87 | 48.50 | 50.37 | 30.26 | 34.20 | 55.88 | 37.41 |
| | 32k | 31.71 | 42.73 | 36.52 | 45.90 | 52.80 | 29.34 | 35.20 | 56.59 | 37.22 |
| 6B Post-norm Depth | 8k | 27.45 | 37.55 | 30.63 | 55.99 | 36.95 | 22.12 | 28.20 | 51.38 | 32.50 |
| | 16k | 25.17 | 37.23 | 28.84 | 56.57 | 36.25 | 21.66 | 27.80 | 52.25 | 31.20 |
| | 24k | 23.01 | 35.13 | 28.41 | 48.38 | 30.78 | 22.43 | 30.20 | 50.59 | 29.07 |
| | 32k | 23.12 | 35.36 | 27.73 | 51.59 | 31.51 | 21.81 | 29.00 | 50.51 | 29.24 |
| 6B Post-norm Width | 8k | 28.98 | 40.16 | 29.44 | 59.39 | 41.54 | 28.11 | 29.60 | 52.88 | 34.57 |
| | 16k | 30.00 | 42.14 | 33.02 | 56.09 | 46.98 | 29.19 | 32.60 | 54.93 | 36.07 |
| | 24k | 32.51 | 42.13 | 34.04 | 48.81 | 50.44 | 30.57 | 34.00 | 54.93 | 37.32 |
| | 32k | 31.73 | 42.63 | 36.35 | 45.81 | 52.78 | 29.65 | 35.00 | 56.20 | 37.18 |

Table 23: Full evaluation results of 17B model pre-training in fig. 1, fig. 11 and fig. 13

| Steps | MMLU | QA_average | Arc_C | BoolQ | Hellaswag | Logiqa | Openbookqa | Winogrande | **Average** |
|---|---|---|---|---|---|---|---|---|---|
| 0k | 25.00 | 33.33 | 25.00 | 50.00 | 25.00 | 25.00 | 25.00 | 50.00 | 29.17 |
| 4k | 29.90 | 40.95 | 35.67 | 53.76 | 47.10 | 25.96 | 32.60 | 50.59 | 35.42 |
| 8k | 34.23 | 45.25 | 41.64 | 50.21 | 57.81 | 27.04 | 37.20 | 57.62 | 39.74 |
| 12k | 40.15 | 48.97 | 46.08 | 54.71 | 65.56 | 28.26 | 38.20 | 61.01 | 44.56 |
| 16k | 43.33 | 50.77 | 47.87 | 61.99 | 64.98 | 28.11 | 39.40 | 62.27 | 47.05 |
| 20k | 46.40 | 51.72 | 49.49 | 59.85 | 66.57 | 28.88 | 41.20 | 64.33 | 49.06 |
| 24k | 48.58 | 53.57 | 51.62 | 64.10 | 67.69 | 30.72 | 41.80 | 65.51 | 51.08 |
| 28k | 49.17 | 54.10 | 52.13 | 65.87 | 68.20 | 29.80 | 40.80 | 67.80 | 51.64 |
| 32k | 51.11 | 54.35 | 52.22 | 66.30 | 69.46 | 28.11 | 40.80 | 69.22 | 52.73 |
| 36k | 51.71 | 55.02 | 55.80 | 67.43 | 69.91 | 26.73 | 41.20 | 69.06 | 53.37 |
| 40k | 53.64 | 55.55 | 56.14 | 66.18 | 71.66 | 28.88 | 41.40 | 69.06 | 54.60 |
| 44k | 55.70 | 56.64 | 55.29 | 69.24 | 73.08 | 29.80 | 42.20 | 70.24 | 56.17 |
| 48k | 56.02 | 56.90 | 56.91 | 64.53 | 73.78 | 29.65 | 43.20 | 73.32 | 56.46 |
| 52k | 57.61 | 57.98 | 57.51 | 70.70 | 73.35 | 29.49 | 43.80 | 73.01 | 57.79 |
| 53k | 57.46 | 58.51 | 57.76 | 71.90 | 73.33 | 32.26 | 44.80 | 71.03 | 57.99 |
| 54k | 58.33 | 58.45 | 59.73 | 68.65 | 73.83 | 30.41 | 45.60 | 72.45 | 58.39 |
| 55k | 57.27 | 57.58 | 57.94 | 68.26 | 72.72 | 29.34 | 44.60 | 72.61 | 57.42 |
| 56k | 58.03 | 58.53 | 57.68 | 72.29 | 74.15 | 29.49 | 44.00 | 73.56 | 58.28 |
| 57k | 57.93 | 58.05 | 58.45 | 68.41 | 74.52 | 30.72 | 43.60 | 72.61 | 57.99 |
| 58k | 58.09 | 58.24 | 57.68 | 70.21 | 73.97 | 31.49 | 43.00 | 73.09 | 58.17 |
| 59k | 58.40 | 58.09 | 56.66 | 69.51 | 74.91 | 31.18 | 43.80 | 72.45 | 58.24 |
| 60k | 58.57 | 58.53 | 58.87 | 70.34 | 74.56 | 31.49 | 43.20 | 72.69 | 58.55 |
| 61k | 58.37 | 58.44 | 57.94 | 71.41 | 74.91 | 29.19 | 44.80 | 72.38 | 58.40 |
| 62k | 58.75 | 58.55 | 58.70 | 70.00 | 74.54 | 29.80 | 45.40 | 72.85 | 58.65 |
| 63k | 59.27 | 58.54 | 58.62 | 71.38 | 74.67 | 29.03 | 43.80 | 73.72 | 58.90 |
| 64k | 59.17 | 58.10 | 57.25 | 70.34 | 75.04 | 29.34 | 44.40 | 72.22 | 58.63 |
| 65k | 59.74 | 57.82 | 57.68 | 67.83 | 75.07 | 29.49 | 43.20 | 73.64 | 58.78 |
| 66k | 58.16 | 58.47 | 57.42 | 71.04 | 74.25 | 30.72 | 43.60 | 73.80 | 58.32 |
| 67k | 59.07 | 59.29 | 58.11 | 75.26 | 74.46 | 30.72 | 44.20 | 73.01 | 59.18 |
| 68k | 59.65 | 58.51 | 58.45 | 72.45 | 74.84 | 29.80 | 42.60 | 72.93 | 59.08 |
| 69k | 59.86 | 58.70 | 58.19 | 72.78 | 74.78 | 30.11 | 43.80 | 72.53 | 59.28 |
| 70k | 60.06 | 59.12 | 58.02 | 73.12 | 75.00 | 29.80 | 44.40 | 74.35 | 59.59 |
| 71k | 59.61 | 59.02 | 59.13 | 72.97 | 74.69 | 29.03 | 44.40 | 73.88 | 59.31 |
| 72k | 59.64 | 59.21 | 58.70 | 74.04 | 75.11 | 30.26 | 43.40 | 73.72 | 59.42 |
| 73k | 60.20 | 59.16 | 59.13 | 71.04 | 75.08 | 31.03 | 45.30 | 73.40 | 59.68 |
| 74k | 59.42 | 59.53 | 57.76 | 73.58 | 75.31 | 32.10 | 44.00 | 74.43 | 59.48 |
| 75k | 60.04 | 58.71 | 58.11 | 69.94 | 75.04 | 32.41 | 44.20 | 72.53 | 59.37 |
| 76k | 59.71 | 59.38 | 59.22 | 72.91 | 74.99 | 31.18 | 44.40 | 73.56 | 59.54 |
| 77k | 60.21 | 58.97 | 58.19 | 73.82 | 75.18 | 30.72 | 43.00 | 72.93 | 59.59 |
| 78k | 60.75 | 59.56 | 58.28 | 75.99 | 75.12 | 30.26 | 44.00 | 73.72 | 60.16 |
| 79k | 60.60 | 59.14 | 58.70 | 71.41 | 75.30 | 31.64 | 44.00 | 73.80 | 59.87 |
| 80k | 59.93 | 59.07 | 59.13 | 71.47 | 75.91 | 30.57 | 44.20 | 73.16 | 59.50 |
| 81k | 60.24 | 59.35 | 58.19 | 75.60 | 75.12 | 29.80 | 43.80 | 73.56 | 59.79 |
| 82k | 60.49 | 59.24 | 57.85 | 73.24 | 75.45 | 31.03 | 44.40 | 73.48 | 59.87 |
| 83k | 61.61 | 60.50 | 59.64 | 77.71 | 76.17 | 30.41 | 44.00 | 75.06 | 61.05 |
| 84k | 61.74 | 60.20 | 59.39 | 75.02 | 76.11 | 30.26 | 44.80 | 75.61 | 60.97 |
| 85k | 61.74 | 60.45 | 59.81 | 75.41 | 76.37 | 31.03 | 44.80 | 75.30 | 61.10 |
| 86k | 62.47 | 59.55 | 58.02 | 72.94 | 75.97 | 30.88 | 44.80 | 74.66 | 61.01 |
| 87k | 62.05 | 60.23 | 58.45 | 74.71 | 76.45 | 31.80 | 45.00 | 74.98 | 61.14 |
| 88k | 62.14 | 60.68 | 60.15 | 75.41 | 75.59 | 32.10 | 45.40 | 75.45 | 61.41 |
| 89k | 61.76 | 59.69 | 60.41 | 73.79 | 76.29 | 29.19 | 43.80 | 74.66 | 60.73 |
| 90k | 62.29 | 60.53 | 60.07 | 73.94 | 76.12 | 31.49 | 45.80 | 75.77 | 61.41 |
| 91k | 62.09 | 59.75 | 58.79 | 74.25 | 75.94 | 31.49 | 43.60 | 74.43 | 60.92 |
| 92k | 62.10 | 60.06 | 59.30 | 74.80 | 76.13 | 30.57 | 44.60 | 74.98 | 61.08 |
| 93k | 62.32 | 60.25 | 59.64 | 74.86 | 76.50 | 30.26 | 45.60 | 74.66 | 61.29 |
| 94k | 62.14 | 59.89 | 59.39 | 74.31 | 76.16 | 31.03 | 44.40 | 74.03 | 61.01 |
| 95k | 61.94 | 59.90 | 60.15 | 72.29 | 76.66 | 31.49 | 44.80 | 74.03 | 60.92 |
| 96k | 62.93 | 60.14 | 59.04 | 78.59 | 76.53 | 29.95 | 42.60 | 74.11 | 61.53 |
| 97k | 62.70 | 60.13 | 60.32 | 72.42 | 76.65 | 30.72 | 45.60 | 75.06 | 61.41 |
| 98k | 62.95 | 60.06 | 59.81 | 74.50 | 76.35 | 31.80 | 44.80 | 73.09 | 61.50 |
| 99k | 62.86 | 60.34 | 59.81 | 74.83 | 76.15 | 31.03 | 44.20 | 76.01 | 61.60 |
| 100k | 62.77 | 59.87 | 60.84 | 72.29 | 76.08 | 29.34 | 45.00 | 75.69 | 61.32 |
| 101k | 62.63 | 60.11 | 59.90 | 75.11 | 76.85 | 29.03 | 44.80 | 74.98 | 61.37 |
| 102k | 62.40 | 60.21 | 59.98 | 73.73 | 77.00 | 30.57 | 45.00 | 74.98 | 61.31 |
| 103k | 62.45 | 59.96 | 59.98 | 72.72 | 76.60 | 29.65 | 44.40 | 76.40 | 61.20 |
| 104k | 62.63 | 60.78 | 60.41 | 74.16 | 76.64 | 31.95 | 46.40 | 75.14 | 61.71 |

Table 24: Full evaluation results of 34B model interleaved growth in fig. 1, fig. 11 and fig. 13

| Steps | MMLU | QA_average | Arc_C | BoolQ | Hellaswag | Logiqa | Openbookqa | Winogrande | **Average** |
|---|---|---|---|---|---|---|---|---|---|
| 60k | 57.96 | 58.05 | 58.10 | 69.02 | 73.93 | 30.72 | 42.60 | 73.93 | 58.01 |
| 61k | 59.61 | 58.64 | 59.04 | 68.47 | 75.86 | 29.95 | 43.40 | 75.14 | 59.13 |
| 62k | 59.54 | 58.60 | 57.85 | 69.48 | 75.46 | 30.11 | 42.80 | 75.88 | 59.07 |
| 63k | 60.25 | 60.27 | 59.47 | 73.24 | 75.84 | 31.80 | 45.60 | 75.69 | 60.26 |
| 64k | 60.51 | 59.66 | 58.45 | 73.15 | 76.18 | 30.88 | 45.60 | 73.72 | 60.09 |
| 65k | 60.57 | 59.29 | 57.85 | 71.22 | 75.92 | 31.18 | 43.80 | 75.77 | 59.93 |
| 66k | 60.19 | 58.90 | 59.30 | 67.80 | 76.06 | 32.41 | 43.40 | 74.43 | 59.55 |
| 67k | 60.54 | 60.04 | 59.40 | 74.25 | 76.18 | 32.41 | 44.20 | 73.80 | 60.29 |
| 68k | 61.50 | 59.40 | 59.47 | 69.82 | 76.19 | 32.72 | 45.20 | 73.01 | 60.45 |
| 69k | 61.42 | 59.53 | 60.67 | 67.49 | 76.65 | 31.95 | 45.60 | 74.82 | 60.48 |
| 70k | 61.67 | 59.66 | 60.49 | 69.11 | 76.74 | 32.26 | 44.60 | 74.74 | 60.66 |
| 71k | 62.08 | 60.36 | 59.98 | 72.57 | 77.09 | 31.64 | 44.40 | 76.48 | 61.22 |
| 72k | 61.97 | 60.34 | 61.01 | 72.05 | 77.03 | 31.95 | 45.00 | 74.98 | 61.15 |
| 73k | 62.21 | 60.29 | 59.90 | 70.80 | 77.06 | 32.26 | 45.00 | 76.72 | 61.25 |
| 74k | 62.74 | 60.99 | 60.67 | 73.15 | 76.97 | 31.18 | 46.40 | 77.58 | 61.87 |
| 75k | 62.85 | 60.67 | 60.15 | 71.41 | 77.33 | 32.10 | 46.00 | 77.03 | 61.76 |
| 76k | 63.13 | 61.19 | 60.49 | 75.60 | 77.42 | 31.03 | 46.80 | 75.77 | 62.16 |
| 77k | 62.92 | 60.41 | 59.81 | 72.05 | 77.09 | 30.57 | 46.80 | 76.16 | 61.67 |
| 78k | 63.52 | 60.05 | 59.90 | 69.88 | 77.47 | 30.11 | 46.20 | 76.72 | 61.78 |
| 79k | 62.65 | 61.27 | 60.84 | 76.18 | 77.11 | 31.49 | 45.20 | 76.80 | 61.96 |

Table 25: Full evaluation results of 34B model stack growth in fig. 11

| Steps | MMLU | QA_average | Arc_C | BoolQ | Hellaswag | Logiqa | Openbookqa | Winogrande | **Average** |
|---|---|---|---|---|---|---|---|---|---|
| 60k | 57.86 | 57.55 | 57.17 | 70.24 | 73.99 | 29.34 | 44.40 | 70.17 | 57.71 |
| 61k | 58.41 | 58.45 | 58.36 | 71.74 | 74.84 | 28.73 | 43.20 | 73.80 | 58.43 |
| 62k | 58.52 | 59.24 | 59.56 | 73.85 | 75.14 | 29.95 | 43.60 | 73.32 | 58.88 |
| 63k | 58.63 | 59.49 | 59.81 | 71.41 | 75.22 | 30.41 | 45.20 | 74.90 | 59.06 |
| 64k | 58.79 | 59.47 | 60.04 | 73.31 | 74.66 | 29.98 | 44.20 | 74.65 | 59.13 |

Table 26: Full evaluation results of 70B model growth in fig. 1 and fig. 13

| Steps | MMLU | QA_average | Arc_C | BoolQ | Hellaswag | Logiqa | Openbookqa | Winogrande | **Average** |
|---|---|---|---|---|---|---|---|---|---|
| 79k | 62.63 | 61.24 | 61.09 | 76.09 | 77.04 | 32.10 | 44.80 | 76.32 | 61.94 |
| 79.5k | 63.59 | 61.86 | 61.09 | 76.42 | 78.15 | 31.64 | 45.40 | 78.45 | 62.72 |
| 80k | 64.49 | 62.29 | 60.58 | 78.41 | 77.77 | 32.41 | 46.20 | 78.37 | 63.39 |
| 80.5k | 64.44 | 62.07 | 62.03 | 74.95 | 77.89 | 32.10 | 47.20 | 78.22 | 63.25 |
| 81k | 65.08 | 62.20 | 62.17 | 78.23 | 77.74 | 31.64 | 46.80 | 76.64 | 63.64 |
| 81.5k | 64.70 | 62.43 | 62.46 | 77.03 | 78.21 | 31.80 | 46.40 | 78.69 | 63.57 |
| 82k | 64.93 | 62.14 | 62.46 | 76.76 | 78.31 | 32.72 | 44.60 | 77.98 | 63.53 |
| 82.5k | 65.55 | 62.10 | 62.54 | 75.41 | 77.96 | 33.49 | 45.40 | 77.82 | 63.83 |
| 83k | 66.06 | 62.52 | 62.71 | 78.47 | 78.08 | 31.64 | 45.20 | 79.01 | 64.29 |
| 83.5k | 65.85 | 61.62 | 61.69 | 75.29 | 78.01 | 30.41 | 45.40 | 78.93 | 63.74 |
| 84k | 65.87 | 62.48 | 63.05 | 77.86 | 78.38 | 31.64 | 45.40 | 78.53 | 64.17 |

# F COMPLEMENTARY EXPERIMENTAL RESULTS

Below we report our complementary experimental results here for more accurate and interpretable visualization for the reviewers.

## F.1 RESULTS FOR ALIGNED ACTIVATED PARAMETERS IN WIDTH/DEPTH GROWTH COMPARISON

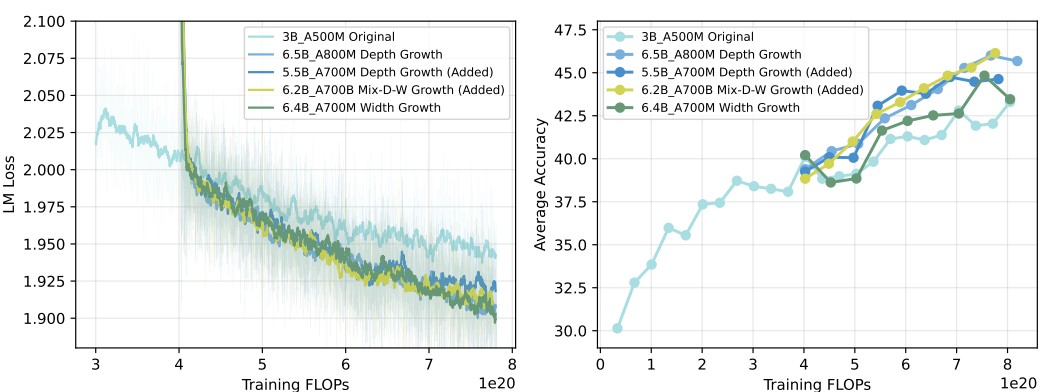

Figure 16: Performance comparison of **aligned** depth growth and width growth. Left: training loss; Right: average downstream task accuracy.

From the results shown here in fig. 16, we can see that the matched-active-parameter depth baseline (5.5B_A0.70B) follows a training curve similar to the original depth-growth experiment (6.5B_A0.79B). This indicates that our conclusion remains the same: depth growth tends to produce better results than width growth, and the effect of width growth appears slower than that of depth growth. These observations are unchanged even when the activated parameters are aligned during training. We also add a preliminary experiment in fig. 16 combining both forms of growth starting from our 3B_A500M model.

## F.2 RESULTS FOR TOPK VARIANTS IN WIDTH GROWTH

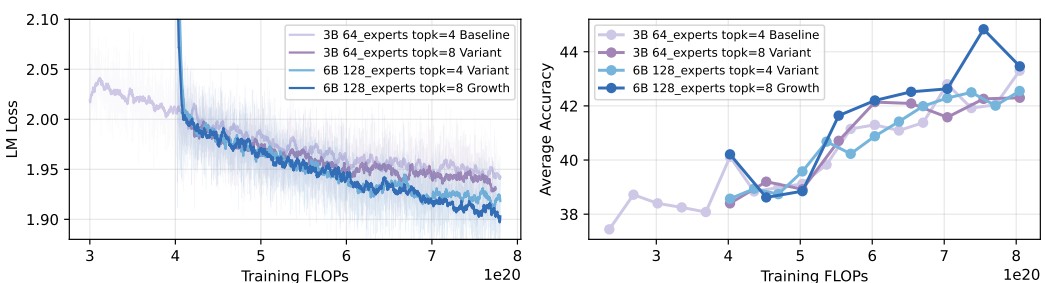

Figure 17: Performance comparison of several topk variants in width growth. Left: training loss; Right: average downstream task accuracy.

Together, these results in fig. 17 reinforce our claim that matching activated parameters or fixed top-k does not alter our main findings: effective model growth requires both architectural expansion and appropriately scaled compute.

## F.3 RESULTS FOR RANDOM BASELINE IN WIDTH GROWTH

The results in fig. 18 show that randomly initializing new experts leads to a large accuracy degradation and prevents further performance improvement. From this observation, we infer that during width growth, knowledge inheritance (through noisy copies) is important for maintaining the base

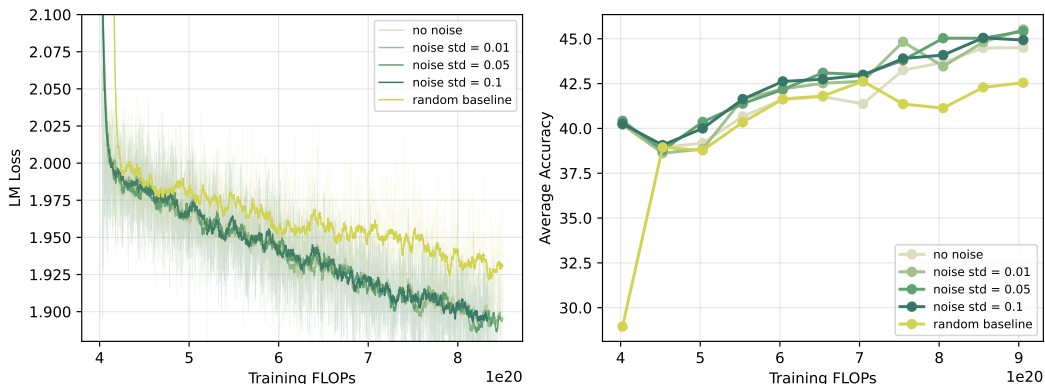

Figure 18: The impact of noise injection scale on width growth performance with random baseline. Left: training loss; Right: average downstream task accuracy.

performance of the grown model. On this basis, adding a small amount of noise helps break symmetry and can further improve downstream performance.

## F.4 RESULTS FOR PROOF OF ORTHOGONALITY

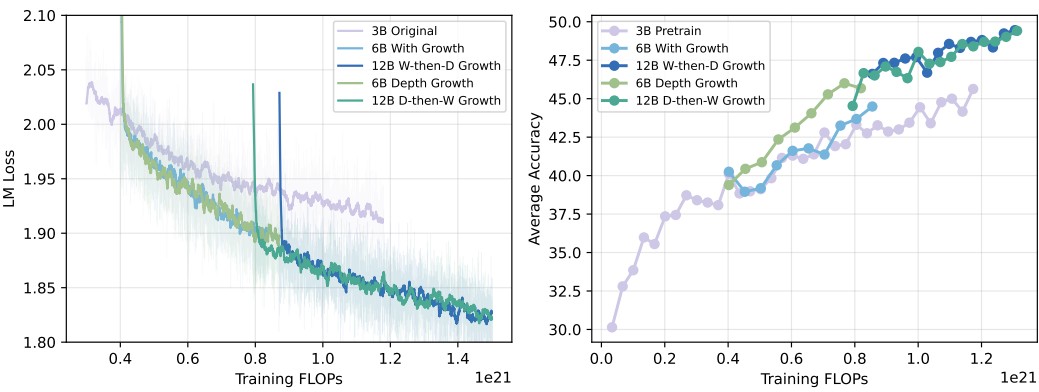

Figure 19: Performance comparison of 3B double growth experiments with two different orders of growth. Left: training loss; Right: average downstream task accuracy.

From these results in fig. 19, we observe that the **depth-then-width** and **width-then-depth** strategies achieve similar performance under the same training FLOPs. Although the route of growth is different, the final performance is equivalent ultimately. This supports the view that the two growth methods are orthogonal. In other words, the order of applying depth and width growth does not affect the final performance.

## F.5 RESULTS FOR EXTRA EXPERIMENTS ON GROWTH FACTOR K

To prove that the ad-hoc choice of growth factor $k$ will not affect our claim that "interposition" method is superior to "stacking" method, we add an additional experiment based on settings in section 3.1 by replacing (k=2) with (k=4), producing a significantly deeper 80-layer model. We report the loss curves and accuracy comparison here:

The results clearly demonstrate that even under (k=4), interposition consistently outperforms stack growth—primarily because interposition better preserves the weight-norm trajectory of the small model, while stack introduces substantial disruptions.

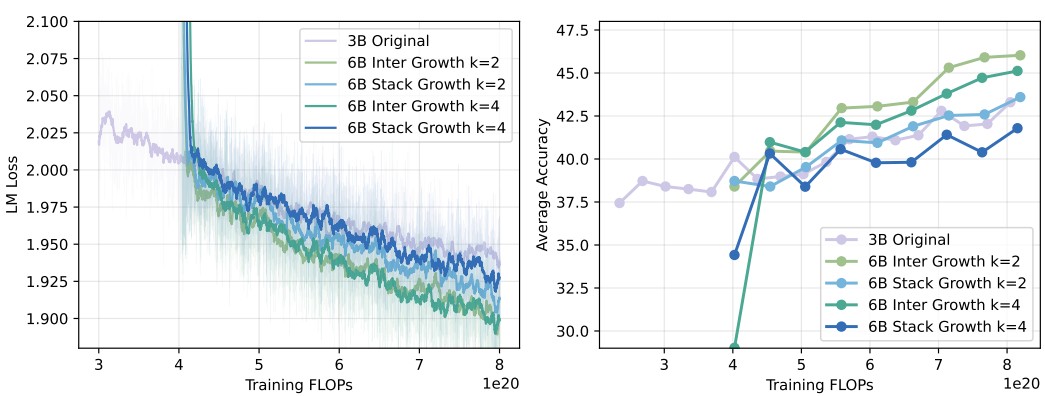

Figure 20: Performance comparison of interposition and stack depth growth strategies under k=2 and k=4. Left: training loss; Right: average downstream task accuracy.

