# OpenReview forum: "Recycling Pretrained Checkpoints: Orthogonal Growth of Mixture-of-Experts for Efficient Large Language Model Pre-Training"
_ICLR.cc/2026/Conference — Submitted to ICLR 2026_

### Official Review · Reviewer_eGL6 · 2025-10-16

**Soundness:** 2
**Presentation:** 2
**Contribution:** 2
**Rating:** 4
**Confidence:** 4

**Summary:**

This paper explores depth and width expansions for upcycling of pretrained MoE LLMs. For depth, each existing transformer layer is duplicated in place rather than appending new blocks. The paper empirically shows this depth expansion better retains layer structure and yields stronger downstream accuracy than stacking. For width expansion, the paper duplicates existing experts and adds Gaussian noise to new experts and router logits to promote divergence. Depth upcycling is compared against width upcycling under matched training compute budgets, demonstrating that depth expansion outperforms expert addition. The paper also investigates the best point during pretraining to upcycle, and compares upcycling to training from scratch and continued pretraining.

**Strengths:**

1- Some of the results, including the empirical comparison between depth and width upcycling, in the context of MoEs are insightful.  Assuming active compute is eventually matched appropriately, the insights from this comparison are informative in large-scale model design and pretraining scaling strategies.

2- The authors make a clear effort to match total training FLOPs across different strategies, which is a strong experimental design choice.

3- The paper experiments with depth and width upcycling at a practically relevant scale. The fact that the upcycled models perform competitively under controlled compute in this scale makes the findings more compelling.

4- The related work section covers most of the key prior work. While a few recent methods could be cited more explicitly (e.g., [1] and [2] in weaknesses), the overall coverage is strong.

5- Even though an LLM is used for polishing writing, the paper is written in a concise and easy to understand style.

**Weaknesses:**

1- Important details are missing or unclear. The number of active parameters for both the base and upcycled MoE models is never stated, only the number of active experts (in the appendix). Figure 1’s pretrain(new) configuration lacks explicit parameter counts, and there is no 70B pretrain result in Figure 11 or the appendix (the tables). These omissions make it difficult to verify the fairness and reproducibility of the reported comparisons. The authors should make sure they include every detail for the readers to understand the experimental settings.

2- The depth and width upcycling comparisons are potentially unfair. The width-scaled model likely activates fewer parameters per token than the depth-upcycled one, which undermines claims of equal compute. The authors should include a matched-active-parameter width baseline, ensuring equal per-token compute. Additionally, a fixed top-k variant is missing, which is important for isolating architectural differences from increased compute. Matching only training FLOPs is not sufficient, since inference efficiency is the key motivation for MoE models.

3- Several plots add little insight or are poorly explained.  Figure 5 evaluates models immediately after upcycling (before continued training) providing no practical signal. Why would you upcycle a model and immediately deploy it? The x-axis labeled training FLOPs is not clear.  Are the points for different model sizes? or different points during training? In general the figure doesn't contribute to the goal of this paper. Also the setting of figure 5 is not clearly explained in the paper. Figure 7 similarly offers an unsurprising conclusion (more training results in better performance?) without new insights. Figure 8 already captures the relevant comparison more cleanly. These figures could be omitted or clarified to improve focus.

4- (Minor) The methodological contributions are incremental. Adding noise to cloned experts has been explored in [1] (as briefly covered in the related work), and duplicating layers in-place was proposed in [2]. The paper’s main difference is applying these known ideas to MoE rather than dense models. Explicitly acknowledging these works and clarifying the scope of genuine novelty would strengthen the positioning.

[1] Drop-Upcycling: Training Sparse Mixture of Experts with Partial Re-initialization, Nakamura et al., 2025

[2] Transformer Layer Injection: A Novel Approach for Efficient Upscaling of Large Language Models, James Vo, 2024

**Questions:**

In addition to the question in the weakness section:

1- Why was a learning rate schedule used instead of a fixed learning rate? Resuming training from different checkpoints along the LR annealing curve (especially near or past the LR floor) could significantly alter training dynamics during upcycling. This seems to be reflected in the paper’s own experiments, where upcycling from late checkpoints under a shared schedule results in degraded performance. If the goal is to isolate the effect of the growth method itself, shouldn’t other variables (like the learning rate) be held fixed or explicitly tuned per checkpoint? Otherwise, it’s unclear whether the observed effects are due to the upcycling start point or artifacts of the LR schedule mismatch.

2- Did the authors try a combination of width and depth scaling at the same time? I understand this expands the design space but a simple experiment that doubles the model size by dividing the added parameters between new experts and new layers could be interesting.

3- Did the authors try training both the upcycled model and the larger model pretrained from scratch to convergence and compare the results?

---

> ### Author Response · Authors · 2025-11-21
> **Official Rebuttal to Reviewer eGL6 by Submission6958 Authors (1/3)**
>
> Thank you for your constructive and insightful comments. We appreciate your helpful suggestions and would also like to clarify a few potential misunderstandings.
>
> **W1: Important details are missing or unclear.**
>
> A1: (1) The activated parameter counts are as follows. For the **3B** model (3,330,979,840 parameters), **500M** (498,595,840) parameters are activated excluding the embedding layer. After growth, the **6B-depth** model (6,457,093,120) activates **792M** (792,325,120) parameters, while the **6B-width** model (6,352,189,440) activates **687M** (687,421,440). For the **17B** model (17,620,395,008), **1.76B** (1,760,764,928) parameters are activated. After growth, the **35B-depth** model (34,831,056,896) activates **3.1B** (3,111,796,736) parameters, and the **35B-width** model (34,537,333,760) activates **2.8B** (2,818,073,600). The final **70B** model (68,664,934,400) activates **5.2B** (5,226,414,080). We apologize for omitting these important model statistics and will update the main text accordingly. We have replaced the '3B' and '17B' model statement with '3B_A500M' and '17B_A1.7B' in the main text and introduce the activated parameters count after model growth in corresponding paragraphs. Detailed total and activated parameter number will be reported in the appendix.
>
> (2) In Figure 1, the parameter count for `pretrain(new)` is 17B, identical to that of `pretrain(old)`. We have made this explicit in the figure. We do not report 70B pre-training results in Figure 11 or the appendix because pre-training a 70B model is beyond our computational budget. It is reasonable for us to use the 17B model for the `pretrain(old)` result rather than the 70B model, because under the training FLOP budget used in Figure 1 (~9×10²¹ FLOPs), a 17B MoE model can be trained on approximately 700B tokens, whereas a 70B model could only be trained on roughly 200B tokens. Such a token count is insufficient for adequately training a model of that scale, and the resulting 70B model would be significantly undertrained.
>
> **W2: The depth and width upcycling comparisons are potentially unfair.**
> A2: We agree that the number of activated parameters per token and inference efficiency are core motivations behind MoE models. In our setting, the width-grown model naturally activates fewer parameters than the depth-grown model because the attention-module parameters are not expanded during width growth. However, our work primarily focuses on the effectiveness of model growth on **training**, and therefore we make substantial efforts to match training FLOPs across different model sizes. **Aligning activated parameters per token or enforcing a fixed top-k baseline does not materially change the main conclusions of our study.**
>
> To support this point, we conducted additional baselines on our 3B_A500M model following your suggestions. We summarize the key results below:
>
> **(1. Matching activated parameters per token.**
>
> We add a matched-active-parameter depth baseline (rather than width), as the two are equivalent for our purpose, and keeping the weight baseline unchanged allows a clean comparison to the fixed top-k variants. The width-grown model is 6.4B_A0.69B, while the depth-grown model is 6.5B_**A0.79B**. We reduce the depth model to 5.5B_**A0.70B** by growing from 20 to 34 layers (instead of 40) while keeping 64 experts (top-k=4), thereby aligning the activated parameters per token. Because we are not doubling the number of layers, we follow the insights from [1,2] and duplicate the central 14 layers. (We note that choosing which layers or experts to copy is an interesting research question but beyond the scope of this paper; here it is required solely to match activated parameters.) We compare this aligned depth baseline with our width growth results:
>
> |Model Setting\Training steps|24k(starting)|26k|28k|30k|32k|34k|36k|38k|40k|
> |-|-|-|-|-|-|-|-|-|-|
> |3B_A500M Pretrain|38.08|40.12|38.84|38.98|39.12|39.83|41.15|41.30|41.09|
> |**5.5B_A700M Depth Growth(added)**|39.25|40.09|40.06|43.08|43.96|43.78|44.75|44.48|44.63|
> |6.5B_A800M Depth Growth|39.39|40.44|40.87|42.35|43.12|44.05|45.28|46.00|45.68|
> |6.4B_A700M Width Growth|40.21|38.62|38.85|41.64|42.20|42.52|42.63|44.83|43.46|
>
> > Loss curves results and accuracy vs. training FLOPs chart have been provided in Figure 16 (Appendix F.1 in the updated pdf)
>
> From the results shown here and the loss curves in updated pdf, we can see that the matched-active-parameter depth baseline (5.5B_A0.70B) follows a training curve similar to the original depth-growth experiment (6.5B_A0.79B). This indicates that our conclusion remains the same: depth growth tends to produce better results than width growth, and the effect of width growth appears slower than that of depth growth. These observations are unchanged even when the activated parameters are aligned during training.

---

> ### Author Response · Authors · 2025-11-21
> **Official Rebuttal to Reviewer eGL6 by Submission6958 Authors (2/3)**
>
> **(2. Fixed top-k variants.**
>
> Starting from the existing 3B_A500M (top-k=4/64) and 6B_A700M (top-k=8/128) width-growth experiments, we add two variants: 6B_A500M (top-k=4/128) and 3B_A700M (top-k=8/64).
>
> |Model Setting\Training steps|24k(starting)|26k|28k|30k|32k|34k|36k|38k|40k|
> |-|-|-|-|-|-|-|-|-|-|
> |3B topk=4 Pretrain|38.08|40.12|38.84|38.98|39.12|39.83|41.15|41.30|41.09|
> |**3B topk=8 Pretrain(added)**|38.40|39.20|38.92|40.71|42.14|42.09|41.58|42.26|42.30|
> |**6B topk=4 Width Growth(added)**|38.57|38.94|38.74|39.58|40.68|40.32|40.88|41.42|41.99|
> |6B topk=8 Width Growth|40.21|38.62|38.85|41.64|42.20|42.52|42.63|44.83|43.46|
> > Loss curves results and accuracy vs. training FLOPs chart have been provided in Figure 17 (Appendix F.2 in the updated pdf)
> - For 6B_A500M, we keep top-k unchanged after growth; this variant basically performs worse than 6B_A700M model, indicating that architectural expansion alone without increased compute does not yield effective training.
> - For 3B_A700M, we donot expand the model parameters and directly increase topk from 4 to 8. This variant initially performs reasonably well, but its subsequent convergence is slow. This suggests that increasing compute alone without architectural expansion limits the model’s ability to continue acquiring knowledge.
>
> Together, these results reinforce our claim that matching activated parameters or fixed top-k does not alter our main findings: effective model growth requires both architectural expansion and appropriately scaled compute.
>
> **W3: Several plots add little insight or are poorly explained.**
>
> A3: We appreciate this concern and would like to clarify the purpose and importance of Figures 5, 7, and 8, as we believe they may have been misunderstood.
>
> (1) The goal of Figure 5 (Section 3.3) is to directly compare the two proposed growth methods—depth growth (Section 3.1) and width growth (Section 3.2) with respect to the *Function Preserving* principle: a grown model's output should remain unchanged **immediately after expansion**. To study this principle, the model must be evaluated immediately after growth, before training resumes. The x-axis of training FLOPs in Figure 5 refers to the FLOPs spent on the original model, i.e., the timing of growth. All training settings match those described in Sections 3.1 and 3.2.
>
> Because *Function Preservation* is widely discussed in prior work on model growth [3–5], we consider this comparison important for demonstrating the advantage of width growth. However, as you noted, this experiment is supplementary relative to the central contributions of the paper. To avoid complexity in the main text, we have moved Figure 5 (right) and its discussion to the Appendix C and only briefly summarize the finding in the main text.
>
> (2) Figures 7 and 8 are essential to our study and cannot be removed. They provide a comprehensive analysis of the **optimal timing** for model growth. The key message of Figure 7 is **not** that “more training yields better performance,” but rather that **greater sunk cost in the base model correlates strongly with improved final performance**. This insight is explicitly discussed in Section 4.1 and was recognized by reviewer wSPh as a major strength of the paper. Figure 8 extends this analysis by studying optimal growth timing under a **fixed total training FLOP budget**, whereas Figure 7 uses a **fixed extra training FLOP budget**. These two conditions represent different practical constraints, and the resulting insights complement each other. Section 4 already explains both figures clearly, and we will ensure this distinction is emphasized.
>
> **W4: The methodological contributions are incremental**
>
> A4: We respectfully clarify the distinctions between our work and the referenced papers. First, the paper [2] you cited is an unreviewed preprint; the publicly available version appears incomplete and contains no experimental results (Section 4 ends after a single line).
>
> For paper [1], the core distinction lies in the underlying methodology. Paper [1] adopts an **"Upcycling"** approach, converting a pre-trained dense model into an MoE model. In contrast, our method expands an already existing MoE model.  **These two methods have a fundamental difference: whether there exists a pre-trained router.** When upcycling, the router does not exist in the original dense model and therefore must be randomly initialized. This inherently introduces substantial randomness, prompting [1] to inject random noise into 50% of the new expert weights to promote expert diversity. But in our setting the model already contains a trained router and trained experts so only a small amount of noise is needed for both the experts and the router to encourage divergence. Injecting excessive noise would instead disrupt the well-trained experts and degrade the final performance. We have updated the Related Work section to explicitly highlight these distinctions and better contextualize our methodological contributions.

---

> ### Author Response · Authors · 2025-11-21
> **Official Rebuttal to Reviewer eGL6 by Submission6958 Authors (3/3)**
>
> Below are our responces to the reviewer's questions:
>
> **Q1: Why was a learning rate schedule used instead of a fixed learning rate?**
>
> A1: A central motivation of our work is to evaluate whether the proposed growth method can effectively **reuse pretrained checkpoints**, many of which have already gone through a full learning rate schedule. Consequently, our experiments (especially those in Figure 7) cover a broad range of intermediate checkpoints, including both the constant-LR phase (8k–64k steps) and the LR-decay phase (72k–96k steps).
>
> Readers interested in isolating the effect of the growth method itself can focus on the 8k–64k region, where the learning rate is constant and the positive correlation between sunk cost and final performance is especially clear. For the LR-decay region (as noted by reviewer 6wRb), growth from very late checkpoints yields smaller gains. This occurs because the grown models resume training with a **larger learning rate** than the decayed LR of their corresponding original checkpoints.
>
> Nevertheless, even under the full scheduler, the **overall positive trend between sunk cost and final performance remains consistent**, supporting our key conclusion.
>
> **Q2: Did the authors try a combination of width and depth scaling at the same time?**
>
> A2: This is indeed an interesting direction. We conducted a preliminary experiment combining both forms of growth starting from our 3B_A500M model. Specifically, we expanded the **number of experts by 1.5×** and the **number of layers by 1.3×**, yielding a 6B-parameter model (6,232,600,064 parameters) with **aligned 700M activated parameters** (709,451,264). The resulting model has **26 layers and 96 experts** (top-k=6).
>
> A key challenge in this setting is determining *which layers and which experts* to duplicate. This is an independent research problem that is beyond the scope of our paper. In this initial attempt, we duplicated the central 6 of the original 20 layers and randomly selected 32 of the 64 experts for copying. We compare this combined-growth variant with the aligned depth and width growth settings discussed in Section 3 (and in our response to W2(1)):
>
> |Model Setting|24k|26k|28k|30k|32k|34k|36k|38k|40k|
> |-|-|-|-|-|-|-|-|-|-|
> |3B_A500M Pretrain               |38.08|40.12|38.84|38.98|39.12|39.83|41.15|41.30|41.09|
> |**6.2B_A700B Mix_D_W Growth (added)**|38.84|39.71|41.00|42.60|43.29|44.10|44.83|45.30|46.14|
> |6.5B_A800M Depth Growth         |39.39|40.44|40.87|42.35|43.12|44.05|45.28|46.00|45.68|
> |**5.5B_A700M Depth Growth (added)**|39.25|40.09|40.06|43.08|43.96|43.78|44.75|44.48|44.63|
> |6.4B_A700M Width Growth         |40.21|38.62|38.85|41.64|42.20|42.52|42.63|44.83|43.46|
>
> > Loss curves results and accuracy vs. training FLOPs chart have been provided in Figure 16 (Appendix F.1 in the updated pdf)
>
> We noticed that this mixed Depth-Width growth method performs comparable or slightly better than simply expanding single dimension, indicating that maintaining the structural shape of the original model (i.e. grow the model in two dimensions at the same time) may be beneficial for final model performance.
>
> **Q3: Did the authors try training both the upcycled model and the larger model pretrained from scratch to convergence and compare the results?**
>
> A3: We summarize our results for the two settings of training a 6B model: growing from a 3B model, or pretraining from scratch. In the model-growth setting, the grown 6B model eventually converges to an accuracy of around 49 (Figure 7), requiring only about 3e20 extra FLOPs. For the 6B from-scratch model, we have currently trained it to an accuracy of around 46 (Table 4) with more than 1.2e21 FLOPs. Although the from-scratch 6B model could be trained further and its final accuracy may eventually exceed that of the growth method, the FLOPs cost would be much larger (nearly 5× or more than that of the growth method). This further proves the main claim of this paper: our growth method can effectively reuse the sunk cost.
>
> Thank you again for taking the time to review our paper. We sincerely appreciate your recognition of our experimental scale, our efforts to match total FLOPs, and the clarity of our writing. We are also grateful for your insightful suggestions, particularly those concerning clearer statements and better alignment regarding activated parameters in MoE models. We hope our responses adequately address your questions and resolve any potential misunderstandings. Please feel free to reach out if you have any further concerns or suggestions.

---

> ### Author Response · Authors · 2025-11-21
> **Official Rebuttal to Reviewer eGL6 by Submission6958 Authors (Important Note and Reference)**
>
> **Important Note**
>
> In all presented data tables above, we're using training steps as the x-axis because in this rebuttal we are constrained to plain text only. We ackowledge that it may introduce unfairness because the total training FLOPs are not perfectly aligned across models, **but it's difficult to present in plain text since each checkpoint corresponds to a different cumulative FLOP count**. For this reason, we report results using training steps (with FLOPs approximately aligned) in the three tables above. At the same time, **we have also included the updated loss and accuracy curves** with a unified FLOP-based x-axis **at the end of revised PDF** for more accurate and interpretable visualization.
>
> **Reference**
>
> > [1] Kim, Sanghoon, et al. "Solar 10.7 b: Scaling large language models with simple yet effective depth up-scaling." Proceedings of the 2024 Conference of the North American Chapter of the Association for Computational Linguistics: Human Language Technologies (Volume 6: Industry Track). 2024.
>
> > [2] Koishekenov, Yeskendir, Aldo Lipani, and Nicola Cancedda. "Encode, Think, Decode: Scaling test-time reasoning with recursive latent thoughts." arXiv preprint arXiv:2510.07358 (2025).
>
> > [3] Evci, Utku, et al. "GradMax: Growing Neural Networks using Gradient Information." International Conference on Learning Representations, 2022.
>
> > [4] Wang, Peihao, et al. "Learning to Grow Pretrained Models for Efficient Transformer Training." The Eleventh International Conference on Learning Representations.
>
> > [5] Yao, Yiqun, et al. "Masked Structural Growth for 2x Faster Language Model Pre-training." The Twelfth International Conference on Learning Representations.

---

### Official Review · Reviewer_wSPh · 2025-10-17

**Soundness:** 3
**Presentation:** 2
**Contribution:** 3
**Rating:** 6
**Confidence:** 4

**Summary:**

This paper addresses the critical issue of high computational costs associated with pretraining large language models (LLMs). The authors propose a framework to "recycle" existing, well-trained checkpoints by expanding them into larger, more capable models, thereby leveraging the "sunk cost" of prior computation. The work specifically focuses on Mixture-of-Experts (MoE) architectures and introduces two orthogonal growth strategies.
The authors conduct experiments showing the superiority of their proposed methods. They investigate the optimal timing for growth, concluding that growing from more converged checkpoints (higher sunk cost) yields better final performance. The framework's scalability is demonstrated by successfully growing a 17B MoE model to a 70B model, which significantly outperforms a from-scratch baseline trained with the same additional compute budget.

**Strengths:**

*   The paper tackles a highly relevant and practical problem. Finding more compute-efficient ways to scale LLMs is crucial for the sustainability and accessibility of AI research.
*   The analysis of growth timing (Section 4) is a key strength. The experiments clearly demonstrate a positive correlation between the amount of prior training and the final performance of the grown model, providing a valuable and actionable insight for practitioners.
*   The paper makes a salient point about the difference in growth strategies for models in early vs. late training stages. The finding that "interposition" is superior for converged models due to preserving structural properties is an interesting and well-argued contribution.
*   The scalability experiment, growing a model from 17B to 70B parameters on a 1-trillion-token scale, is impressive and provides strong evidence for the practical viability of the proposed framework.

**Weaknesses:**

2.  **Inadequate Ablation Studies for Key Design Choices:**
    The framework's effectiveness relies on several critical design decisions that are not fully justified through ablation studies.
    *   **Width Growth Initialization:** The paper demonstrates that "copying with noise" is superior to noiseless copying. However, a much stronger and more intuitive baseline is to **randomly initialize the new experts** while retaining the original ones. Comparing against this baseline is necessary to determine whether the advantage comes from knowledge inheritance (via noisy copies) or simply from effective symmetry-breaking.
    *   **Orthogonality and Order of Growth:** The paper claims depth and width growth are "orthogonal" but only demonstrates a fixed "depth-then-width" sequence in the large-scale experiments. This does not sufficiently prove orthogonality, as the order of operations could matter in practice. An experiment testing the reverse order ("width-then-depth") is needed to validate this claim and explore potential performance differences.

**Questions:**

1.  You convincingly argue that the "interposition" method is superior for **converged** models, contrasting this with prior work like [4] that applies "stacking" in earlier training phases. Could you elaborate on where this tipping point might be? Is there a stage in early-to-mid training where "stacking" performs comparably or even better?

2.  For width growth, you used Gaussian noise to break symmetry. Did you experiment with other noise types or alternative symmetry-breaking techniques? Specifically, how does the proposed "copy-and-noise" approach compare empirically to the baseline of randomly initializing the new experts?

3.  In your 70B experiment, you applied a "depth-then-width" strategy. What is the intuition behind this order? Given that depth growth enhances functional complexity while width growth expands model capacity, could the sequence of these operations fundamentally impact the final model's capabilities or its training dynamics?


[1] Liu D, Wang Z, Wang B, et al. Checkpoint merging via bayesian optimization in llm pretraining[J]. arXiv preprint arXiv:2403.19390, 2024.

[2] Peihao Wang, Rameswar Panda, Lucas Torroba Hennigen, Philip Greengard, Leonid Karlinsky, Rogerio Feris, David Daniel Cox, Zhangyang Wang, and Yoon Kim. Learning to grow pretrained models for efficient transformer training. arXiv preprint arXiv:2303.00980, 2023a.

[3] Peihao Wang, Rameswar Panda, and Zhangyang Wang. Data efficient neural scaling law via model reusing. In International Conference on Machine Learning, pp. 36193–36204. PMLR, 2023b.

[4] Du W, Luo T, Qiu Z, et al. Stacking your transformers: A closer look at model growth for efficient llm pre-training[J]. Advances in Neural Information Processing Systems, 2024, 37: 10491-10540.

---

> ### Author Response · Authors · 2025-11-21
> **Official Rebuttal to Reviewer wSPh by Submission6958 Authors (1/2)**
>
> Thank you for your thorough and constructive feedback on our work and for recognizing the strengths of our contributions. Below we address your concerns and questions:
>
> **W2: Inadequate Ablation Studies for Key Design Choices**
>
> A2: We thank the reviewer for the insightful suggestions and acknowledge the importance of adding this ablation study. Below we provide the complementary experiment and analyze the conclusions.
>
> **(1) Width Growth Initialization.** We add a random-initialization baseline for width growth on our 3B_A500M model, where all new experts are randomly initialized while the original ones are retained. The router is also fully randomly initialized. We compare this baseline with our normal width growth experiments both with and without adding noise.
>
>
> |Model Setting\Training steps|24k(starting)|26k|28k|30k|32k|34k|36k|38k|40k|42k|44k|
> |-|-|-|-|-|-|-|-|-|-|-|-|
> |6B **Random** Width Growth **(added)**|28.95|38.94|38.78|40.34|41.64|41.81|42.63|41.36|41.13|42.29|42.55|
> |6B Width Growth no noise     |40.25|38.95|39.18|40.67|41.61|41.77|41.37| 43.25|43.68|44.49|44.50|
> |6B Width Growth noise=0.01   |40.21|38.62|38.85|41.64|42.20|42.52|42.63|44.83|43.46|44.81|45.52|
> |6B Width Growth noise=0.05   |40.42|38.78|40.36|41.38|42.17|43.10|43.00| 43.80|45.03|45.03|45.42|
> |6B Width Growth noise=0.1    |40.25|39.07|40.00|41.63|42.62|42.74|42.97| 43.90|44.09|45.05|44.93|
>
> > Loss curves results and accuracy vs. training FLOPs chart have been provided in Figure 18 (Appendix F.3 in the updated pdf)
>
> The results show that randomly initializing new experts leads to a large accuracy degradation and prevents further performance improvement. From this observation, we infer that during width growth, knowledge inheritance (through noisy copies) is important for maintaining the base performance of the grown model. On this basis, adding a small amount of noise helps break symmetry and can further improve downstream performance.
>
> **(2) Orthogonality and Order of Growth.** We acknowledge the importance of adding more experiments to examine the orthogonality between depth growth and width growth. We conduct two additional experiments on our 3B_A500M model (20 layers, 64 experts with 4 activated): a *depth-then-width* strategy and a *width-then-depth* strategy, both yielding a final 12B_A1.5B model (40 layers, 128 experts with 8 activated). We first train the base 3B model with 4e20 FLOPs, then train 2e20 FLOPs for the first growth stage, and finally 6e20 FLOPs for the second growth stage. The accuracy results are shown below:
>
> |Model Setting\Training FLOPs (e20)|8.6(starting)|9.1|9.4|9.7|10|10.3|10.6|10.9|11.2|11.5|11.8|12.1|12.4|12.7|
> |-|-|-|-|-|-|-|-|-|-|-|-|-|-|-|
> |12B Depth-then-Width|46.61|47.32|47.33|47.61|47.70|46.69|47.98|48.56|48.30| 48.69|48.81|48.33|49.24|49.45|
> |12B Width-then-Depth|46.51|47.09|46.74|46.33|48.04|47.27|47.38|47.72|48.55| 48.40|48.69|48.71|49.02|49.41|
>
> > Loss curves results and accuracy vs. training FLOPs chart have been provided in Figure 19 (Appendix F.4 in the updated pdf)
>
> From these results, we observe that the *depth-then-width* and *width-then-depth* strategies achieve similar performance under the same training FLOPs. Although the route of growth is different, the final performance is equivalent ultimately. This supports the view that the two growth methods are orthogonal. In other words, the order of applying depth and width growth does not affect the final performance.
>
> **Note**
>
> For loss curves and better visualization of the two data tables shown above, the updated loss and accuracy curves with a unified FLOP-based x-axis have also been updated at the end of the revised PDF.

---

> ### Author Response · Authors · 2025-11-21
> **Official Rebuttal to Reviewer wSPh by Submission6958 Authors (2/2)**
>
> **Q1: Tipping point for training status**
>
> A1: This is a very interesting and valuable question related to a scaling-law-like phenomenon. Our claim that the “interposition” method is more suitable for converged models is based on the observation that well-trained models exhibit a distinct pattern in their layer-wise weight norms, and the interposition method can better preserve this functional structure. **Therefore, the tipping point at which interposition begins to outperform the stack method may correspond to the point in training when this distinct layer-wise weight-norm pattern emerges.** Before reaching this tipping point, the stack method may perform better.
>
> For example, in both our paper and your referenced work [4], the total training FLOPs for the ablation setting (3B_A500M → 6B_A1B in our paper and 440M → 1.1B in their paper) are 8e20 FLOPs. Regarding the timing of growth, [4] allocates the FLOPs ratio before vs. after growth at approximately 1:39 and observes that stacking performs better. At this timing, the model is clearly not converged. In contrast, we choose a 20:20 ratio in our experiments and observe a clear upward weight-norm pattern in the checkpoint at the growth point. We also examine the model state at a timing comparable to the 1:39 ratio in our setting and find that the weight-norm distribution is still very random and does not show the upward trend. This observation further supports our hypothesis.
>
> **Q2: Did you experiment with other noise types or alternative symmetry-breaking techniques?**
>
> A2: In our experiments, our initial motivation was to use small Gaussian noise simply to break symmetry and encourage the grown experts to diverge, so we did not explore other noise types, as we expect different noise distributions would not substantially change the symmetry-breaking effect. Regarding alternative symmetry-breaking techniques, as shown in response A2 to W2(1), the baseline with randomly initialized new experts performs worse than the “copy-add-noise” method. This result suggests that symmetry breaking mainly acts as a performance-boosting technique, while the fundamental base performance comes from knowledge inheritance through copied experts.
>
> **Q3: What is the intuition behind the order of "depth-then-width" strategy?**
>
> A3: Our two proposed methods (depth growth and width growth) are structurally orthogonal, so we hypothesize that the order of growth should not affect the final performance. Therefore, in our 70B experiment, we arbitrarily selected the “depth-then-width” strategy. As shown in response A2 to W2(2), the ablation study on the 12B model comparing the “depth-then-width” and “width-then-depth” strategies shows that the order of growth indeed does not affect the final model performance.
>
> ---
>
> We thank the reviewer again for recognizing our work and for the constructive feedback, which has helped us strengthen our ablation study. We hope our responses have addressed your questions, and we would be happy to discuss further if you have any additional suggestions.

---

### Official Review · Reviewer_6wRb · 2025-10-28

**Soundness:** 2
**Presentation:** 3
**Contribution:** 3
**Rating:** 2
**Confidence:** 3

**Summary:**

The paper utilizes two main techniques to upcycle a pretrained (MoE) model to increase the capacity of the model:
- interpositional layer copying for depth growth
- expert duplication for width growth
They claim that utilizing these methods leverage the sunk cost of pretrained models to accelerate learning.
These methods first modify the model architecture, before continual pretraining on the model.

The depth-growth method is inspired by the fact that the norm of each layer is more aligned to each other when using the interpositional stacking method.
The width growth method is by duplicating expert and adding some noises to it.

The largest concern is that the proposed separate methods have been shown to be ineffective in previous work, and it is not clear why they work in the paper's proposal.

**Strengths:**

- The experiments conducted are at fairly large scale: 70B model with over 1T training tokens.
- The paper is quite easy to follow with clear organization, with ablation studies to support their arguments (with caveats; see below)

**Weaknesses:**

- The utilized methods for depth growth and width growth are well-known and have been shown to be less effective. It is unclear what differences are made in the paper that make them work:
  - depth growth: [2405.15319] show that naive stacking works best; while the paper claims that it is suboptimal for converged models, the argument is not convincing, as no detailed comparison is made showing under what condition (at what sunk cost, budget, model size etc.) stacking works best vs interpositional stacking. Also, models in  [2405.15319] are trained with over the chinchilla-suggested token number and are quite overtrained.
  - width growth: In [2212.05055,2406.06563,2409.02060,2502.03009], it was shown that adding gaussian noises does not lead to performance improvement. [2502.03009] further shows that it is more difficult to upcycle a model from an overtrained base model. These results contradict the claim of the paper.
- When studying the impact of sunk cost, the paper should also use checkpoints with LR fully decayed, instead of intermediate checkpoints from a single run. This is more useful as in practice, one upcycle a model which has finish training (LR fully decayed).
- Design choices are quite ad-hoc, e.g., increasing the depth/width by a factor of 2 (not 3, 4, etc).

**Questions:**

Please address the weaknesses mentioned above and clarify if there is any misunderstanding.

---

> ### Author Response · Authors · 2025-11-13
> **Official Rebuttal to Reviewer 6wRb by Submission6958 Authors (1/2)**
>
> Thank you for your feedback on our work. We appreciate the opportunity to discuss your concerns and clarify our approach, which we believe has been misinterpreted.
>
> **W1: Growth methods are useless?**
>
> **A1:** **We believe there is a misunderstanding regarding the effectiveness of our proposed growth methodology.** We believe these contradictions stem from crucial difference in experimental settings (model convergence state) and inherent methodology (MoE "Expansion" vs. Dense-to-MoE "Upcycling"). We will provide a detailed explanation of each component to clarify its rationale and performance.
>
> ---
>
> **W1(1) - Depth growth**
>
> First, our claim that interpositional layer copying is superior to naïve stacking is based on two factors: observations of distinct layer-wise weight norms in pre-trained models and end-to-end experiments on our converged 3B and 17B MoE models.
>
> Regarding the different results in [2405.15139], their ablation studies (Section 5.1 and Appendix H) were conducted on a **400M (6 layers) Llama model grown to 1.1B (24 layers)**. The 400M model was trained for **10B tokens** and the 1.1B model for **97.5B tokens**, with their results favoring naïve stacking.
>
> However, we argue this is due to a critical difference in experimental setup. In [2405.15139], despite their models being significantly overtrained (exceeding Chinchilla-suggested token counts), the FLOPs budget ratio *before* growth to *after* growth was approximately **1:39**. This implies **the small model was clearly under-trained and not converged at the point of growth**.
>
> In contrast, our experiments (Section 3.1) on the 3B(A500M) $\to$ 6B(A1B) model used a before-to-after FLOPs ratio of nearly **1:1** (Figure 3). Notably, the *total* FLOPs budget (8e20) are comparable in both our work (Figure 3) and [2405.15139] (Figure 33) for this scale. Furthermore, in our 17B(A2B) $\to$ 35B(A4B) setting (Figure 9), this ratio was approximately **5:1**, with a total FLOPs budget of 8e21.
>
> It is clear that under our settings, the small model is significantly more converged prior to growth than in the settings used by [2405.15139]. To clarify this distinction, we will add these comparative statistics to our paper to detail the convergence state of the base model.
>
> ---
>
> **W1(2) - Width growth**
>
> First, we wish to note that the referenced papers [2212.05055, 2406.06563, 2409.02060, 2502.03009] all relate to **Upcycling**—converting a pre-trained dense model into an MoE model. In contrast, our method **expands** an existing MoE model. **These two methods have a fundamental difference: whether there exists a pre-trained router.**
>
> In the Upcycling papers, a new router must be randomly initialized since original dense model doesn't have a router. **This action inherently introduces significant randomness.** Consequently, adding further noise (e.g., the 50% Gaussian noise to experts in [2212.05055, 2409.02060]) may not yield better performance.
>
> In our setting, however, we duplicate **both** the experts **and the pre-trained router**. Without added noise, the new MoE's output would be identical to the old one, preventing divergence. Therefore, we add a very small noise (mean 0, std $0.01 \times \sigma_{orig}$, where $\sigma_{orig}$ is the original weights' standard deviation) to both experts and the router, encouraging them to diverge during continued training. Experiment results in Figure 4 clearly support our perspective.
>
> Furthermore, regarding the scaling law results in [2502.03009]:
>
> * This scaling law was derived from the *upcycling* setting, not our MoE *expansion* setting, so it may not be directly applicable.
> * Even if the scaling law were universal, the claim in [2502.03009] is that increasing D1 (sunk cost) reduces initial loss but results in **slower training progress** with D2 (upcycled MoE tokens), or in other words making training more difficult, **but not harmful**.
> * In our paper we claim that more sunk cost leads to better *final* performance given the same *extra* FLOPs. This positive correlation is not contradictary to [2502.03009].
>
> In fact, our findings in Section 4.2 are consistent with this insight. We show that under the same total FLOPs budget, growing from a late checkpoint (i.e., the LR decay phase) yields a smaller performance gain than growing from a middle checkpoint (Figure 8 and Table 2). This suggests that "one should allocate additional FLOPS at least on the same order of magnitude as the sunk cost in order to achieve performance comparable to pre-training under the same total FLOPs" (Lines 367-369). This conclusion actually aligns with the insights from [2502.03009].

---

> ### Author Response · Authors · 2025-11-13
> **Official Rebuttal to Reviewer 6wRb by Submission6958 Authors (2/2)**
>
> **W2: Effect on LR fully decayed model**
>
> **A2:** We did include this scenario in our experiments. The **96k checkpoint**, referenced in Figure 7 and Table 1, represents a model where the learning rate has fully decayed to its minimum value (as shown in Figure 6).
>
> Our final four checkpoints (72k, 80k, 88k, and 96k) were chosen to fully cover the entire learning rate decay phase, from its start (72k) to its completion (96k). We did not emphasize the 96k checkpoint separately because the positive trend regarding the impact of sunk cost remained consistent across this phase.
>
> As shown in Figure 7 and Table 1, the model grown from this fully decayed (96k) checkpoint exhibits a minor performance degradation compared to the model grown at 88k steps. However, it still significantly outperforms the models grown from all earlier checkpoints.
>
> **W3: Design choices are quite ad-hoc**
>
> **A3:** We choose the ad-hoc growth factor $k=2$ following established precedent in related work on model growth and upcycling. Using a fixed, ad-hoc growth factor is standard practice in this field.
>
> For example, several of the papers you referenced also use this approach:
> * [2405.15139] uses a growth factor of $k=4$.
> * [2406.06563] and [2502.03009] use $k=8$ (for the number of upcycled experts).
> * [2212.05055] is an exception as the growth factor itself is their primary research focus.
> * [2409.02060] does not use upcycling.
>
> This practice is also common in other key papers on model growth:
> * Paper [1] uses fixed $k=4$ (2x width and 2x depth).
> * Paper [2] uses fixed $k=1.25$ (32 to 40 layers).
> * Paper [3] uses fixed $k=2$ (hidden\_size $6384 \to 12768$).
> * Paper [4] uses fixed $k=1.5$ (32 to 48 layers).
>
> ---
>
> We hope these clarifications address your concerns and resolve any misunderstandings. If you have other questions or points of confusion, please contact us again, and we will gladly provide a response.
>
> ---
> > [1] Shen, Sheng, et al. "Staged training for transformer language models." International Conference on Machine Learning. PMLR, 2022.
>
> > [2] Wu, Chengyue, et al. "LLaMA Pro: Progressive LLaMA with Block Expansion." Proceedings of the 62nd Annual Meeting of the Association for Computational Linguistics (Volume 1: Long Papers). 2024.
>
> > [3] Wang, Yite, et al. "LEMON: Lossless model expansion." The Twelfth International Conference on Learning Representations.
>
> > [4] Kim, Sanghoon, et al. "Solar 10.7 b: Scaling large language models with simple yet effective depth up-scaling." Proceedings of the 2024 Conference of the North American Chapter of the Association for Computational Linguistics: Human Language Technologies (Volume 6: Industry Track). 2024.

---

> ### Comment · Reviewer_6wRb · 2025-11-22
>
> Thanks for the response and clarification regarding the distinction between MoE "Expansion" (current work) and Dense-to-MoE "Upcycling" (cited prior work). I understand the fundamental difference between them which could justify the addition of noise in contrast to prior work.
>
> However there remain questions and concerns unresolved. Please clarify again if I misunderstood anything.
> My biggest concern is that there isn’t sufficient investigation on at what situation the proposed method works, detailed below.
>
> 1. Your argument for the superiority of the interposition method rests on the premise that the base model must be sufficiently converged or "overtrained," unlike the settings in prior work [2405.15319]. To validate this claim and make your method generally useful, the paper should quantitatively define the boundary conditions for this superiority.
> - E.g., ablation study is needed to systematically vary the pre-growth FLOPs ratio (e.g., from 1:39 up to 5:1 and beyond) to establish at what exact point (e.g., ratio of pre-to-post growth FLOPs, or loss/accuracy state) the interposition method reliably outperforms stacking. Without this, your conclusion is only an empirical observation under your specific, more converged model.
>
> 2. My concern on ad hoc design is also similar: Does the superiority of interposition persist when the number of layers is significantly larger, potentially disrupting the learned weight norm trend to a greater degree? The same goes to width expansion. Since this work largely utilizes known growth techniques in the literature to grow the model, one should systematically characterize when and how much this particular (combination of known techniques) method works, to add scientific value and transferability to the community.
>
> 3. Sunk Cost Correlation: only the last checkpoint you investigated is fully decayed, and others are intermediate checkpoints with different LR, am I correct?
> - Most publicly released "fully trained" models would be checkpoints saved after the full LR decay. If we are to grow an off-the-shelf converged model, it would likely be one trained for a greater number of tokens and a fully decayed LR, which is what I meant by a "full decayed model" checkpoint.
> - to truly model the practical scenario of recycling an off-the-shelf model (i.e., a "fully converged" or "fully decayed" model), you should compare growth experiments starting from models trained with varied total tokens (sunk cost), where each base model has completed its full, intended learning rate decay.

---

> > ### Author Response · Authors · 2025-11-28
> > **Official 2nd round Rebuttal to Reviewer 6wRb (1/3)**
> >
> > We thank the reviewer for the follow-up response. We are glad that our initial reply addressed part of your concerns, and we appreciate the opportunity to further clarify and discuss your additional questions.
> >
> > ---
> >
> > **Q1: To validate this claim (“interposition” > “stack”), the paper should quantitatively define the boundary conditions for this superiority.**
> >
> > **A1:** We fully agree on the importance of quantitatively characterizing the convergence boundary at which *interposition* begins to outperform *stack*. To this end, we conducted a systematic study on the **3B_A500M** model to determine this boundary (i.e. at which certain degree of model convergence when the two methods performs differently) .
> >
> > Intuitively, we use the **amount of training FLOPs** as a measure of training progress. According to the **Chinchilla Scaling Law** [1], the compute-optimal ratio between model size (N) and token count (D) is approximately (D = 20N), and the corresponding total compute can be estimated as (6ND) (where the factor 6 follows from the empirical rule of forward + backward computation).
> > For MoE models, the situation is more complicated, but a reliable approximation is to base the compute on the **number of activated parameters**, Na = 500M, instead of the total parameter count [2]. For our **3B_A500M** model, the resulting compute-optimal FLOPs (F_c) is approximately:
> >
> >  $ F_c \approx 6 \cdot N_a \cdot (20N_a) \approx 3 \times 10^{19} $
> >
> > In practice, small models are commonly **overtrained** well beyond this compute-optimal value.
> >
> > Using the accumulated FLOPs of the 3B_A500M model as the indicator, we examined checkpoints at various training stages. At each selected checkpoint, we applied either *stack* or *interpositional* layer growth, then continued training for a fixed number of steps and compared their performance. We selected several checkpoints roughly corresponding to integer multiples of (F_c) as growth points. The results are summarized in the table below.
> >
> > > timing: 2k (3.35e19, ~1*Fc)   "Stack" better
> >
> > |Training steps|2k|4k|6k|8k|10k|12k|14k|16k|
> > |-|-|-|-|-|-|-|-|-|
> > |Stack Growth          |  29.93  |  31.91  |**35.38**|**37.48**|**38.67**|**38.77**|**39.78**|  40.08  |
> > |Interpositional Growth|  30.12  |**33.15**|  35.27  |  36.29  |  36.56  |  37.76  |  39.51  |**40.72**|
> >
> > > timing: 4k (7e19, ~2*Fc)   Similar Performance
> >
> > |Training steps|4k|6k|8k|10k|12k|14k|16k|18k|
> > |-|-|-|-|-|-|-|-|-|
> > |Stack Growth          |**32.57**|  33.77  |**36.08**|  37.56  |  38.55  |  39.06  |  39.96  |  38.88  |
> > |Interpositional Growth|  31.72  |  33.87  |  35.92  |**37.98**|  38.56  |**40.09**|**40.58**|**40.64**|
> >
> > > timing: 6k (1e20, ~3*Fc)   "Interpositional" better
> >
> > |Training steps|6k|8k|10k|12k|14k|16k|18k|20k|
> > |-|-|-|-|-|-|-|-|-|
> > |Stack Growth          |**34.14**|**35.56**|  36.32  |  37.28  |  37.47  |  38.64  |  39.38  |  38.89  |
> > |Interpositional Growth|  32.97  |  34.67  |**37.71**|**38.50**|**40.56**|**40.35**|**41.81**|**41.38**|
> >
> > > timing: 8k (1.3e20, ~4*Fc)   "Interpositional" better
> >
> > |Training steps|8k|10k|12k|14k|16k|18k|20k|22k|
> > |-|-|-|-|-|-|-|-|-|
> > |Stack Growth          |**35.94**|  35.26  |  38.33  |  38.73  |  39.50  |  39.70  |  39.50  |  40.12  |
> > |Interpositional Growth|  34.06  |  **35.47**  |  38.40  |**40.06**|**41.25**|**41.42**|**41.89**|**41.53**|
> >
> > (full results including loss-curve comparisons are provided in **Figure 4** of **Section 3.1** in the revised PDF)
> >
> > We observe that the critical point at which the two growth methods begin to diverge emerges at approximately **2× the Chinchilla-optimal FLOPs (F_c)**. This suggests that once the total training FLOPs exceed **twice F_c**, the **interpositional** method should be preferred over the **stack** method, indicating that the model has reached a well-converged state.
> >
> > For comparison, in *[2405.15319]*, the authors apply layer copying at
> >
> > $ \frac{1}{40} \times 8\times10^{20} \approx 2\times10^{19} $  FLOPs
> >
> > and for their **440M dense model**, the corresponding Chinchilla-optimal compute is
> >
> > $ F_c = 6N(20N) = 2.32\times10^{19} $ FLOPs
> >
> > which is very close to **1×F_c**. Their finding that *stack* performs better than *interposition* under continued training is therefore **fully consistent with** our own results at roughly **1×F_c**.
> >
> > In addition, **to further connect this observation with our findings on weight norms**, we examined the weight-norm distributions of the checkpoints used in the four experiments above to further prove our motivation. The corresponding plots have been updated in the revised PDF (Figure 5). (Please continue to the 2nd part)
> >
> > > [1] Hoffmann, Jordan, et al. "Training compute-optimal large language models." arXiv preprint arXiv:2203.15556 (2022).
> >
> > > [2] Fedus, William, Barret Zoph, and Noam Shazeer. "Switch transformers: Scaling to trillion parameter models with simple and efficient sparsity." Journal of Machine Learning Research 23.120 (2022): 1-39.

---

> > ### Author Response · Authors · 2025-11-28
> > **Official 2nd round Rebuttal to Reviewer 6wRb (2/3)**
> >
> > **Continued responce to Q1**
> >
> > Figure 5 in revised pdf shows that, early in training, all weights are initialized from an i.i.d. Gaussian distribution with a fixed standard deviation (σ = 0.02), leading to uniform layer-wise norms. As training progresses, the weight-norm distribution develops a clear upward trend. Around **4k steps**, a stable **layer-wise increasing pattern** emerges, and in subsequent training the norms continue to grow while maintaining this structure.
> >
> > Therefore, we believe that **the emergence of this characteristic increasing pattern across layers marks the boundary at which interpositional growth begins to outperform stack growth**. We have updated this part in the main text of the paper and thanks the reviewer very much for this precious suggestion.
> >
> > ---
> >
> > **Q2: Concern on ad-hoc design and the combination of known growth techniques**
> >
> > **A2:** In our paper, the claimed advantage of *interposition* is grounded in the observed trend of weight norms during growth. **Although setting (k=2) is indeed an ad-hoc choice, it does not affect our central conclusion: *interposition* preserves the increasing pattern of weight norms, whereas *stack* disrupts it.** In fact, with larger (k) values that yield deeper grown models, the weight norms under interposition remain at least non-decreasing, while stack growth frequently exhibits repeated rise-then-fall behaviors, resulting in a more chaotic pattern. For width growth, since expert copying is independent of layerwise growth order, we similarly believe that the ad-hoc choice of (k) does not influence our claim.
> >
> > **To further support this point, we added an additional experiment based on settings in Section 3.1 by replacing (k=2) with (k=4), producing a significantly deeper 80-layer model.** We applied both interposition and stack growth to this model and report the continued-training results below. The results clearly demonstrate that even under (k=4), interposition consistently outperforms stack growth—primarily because interposition better preserves the weight-norm trajectory of the small model, while stack introduces substantial disruptions.
> >
> > | Model Setting \ Training Steps | 24k (starting) | 25k       | 26k       | 27k       | 28k       | 29k       | 30k       | 31k       |
> > | ----- | ----- | ---- | ---- | --- | ---- | - | - | - |
> > | 6B, (k=4), interposition       | 29.02          | **40.98** | **40.40** | **42.13** | **41.99** | **42.81** | **43.80** | **44.72** |
> > | 6B, (k=4), stack               | **34.42**      | 40.32     | 38.39     | 40.57     | 39.78     | 39.81     | 41.41     | 40.39     |
> >
> > We have also updated the loss-curve comparison corresponding to this table in the revised PDF (Figure 20 in Appendix F.5), which further confirms that interposition outperforms stack.
> >
> > We thank the reviewer for raising this concern. We agree that clarifying the connection between our method and existing growth techniques is important for its relevance to the research community. Accordingly, we have revised Section 3 to more clearly restate our key observations and motivations and incorporated additional experiments to substantiate these points.
> >
> > ---
> >
> > **Q3: Sunk Cost Correlation**
> >
> > **A3:** To answer the first part of the question: **yes**, in our investigation of sunk-cost correlation, only the *final* checkpoint is trained under a fully decayed learning rate (LR). All intermediate checkpoints are taken from earlier phases of training and therefore correspond to different LR values, many of which are at the maximum constant LR. Our motivation is to examine whether the positive relationship between sunk cost and final performance persists across different LR phases. This design aims to provide insight for researchers and practitioners who accumulate many intermediate checkpoints during large-scale LLM training.
> >
> > We fully acknowledge the practical importance of checkpoints in real-world scenarios, especially given that most high-performing open-source models have already passed through the complete LR decay phase. Directly applying our training-based sunk-cost analysis to such models would be ideal. However, attempting continued training on these models poses two challenges:
> >
> > 1. The proprietary datasets used to train these models are unavailable, and our collected data is very likely of lower quality, which can lead to *performance degradation* during continued training, as also noted in [3]
> > 2. The reviewer’s suggestion to emulate practical settings by training many models, each with varied sunk cost and fully decayed LRs, is entirely reasonable, but the design space and computational requirements are prohibitively large for us.
> >
> > As a compromise, we performed an intermediate trade-off experiment. (Please continue to the 3rd part)
> >
> > > [3] Bae, Sangmin, et al. "Relaxed Recursive Transformers: Effective Parameter Sharing with Layer-wise LoRA." The Thirteenth International Conference on Learning Representations.

---

> > ### Author Response · Authors · 2025-11-28
> > **Official 2nd round Rebuttal to Reviewer 6wRb (3/3)**
> >
> > **Continued responce to Q3**
> >
> > As a compromise, we performed an intermediate trade-off experiment: we applied our growth method directly to the weights of several powerful open-source MoE models **(e.g., Qwen and DeepSeek)** and evaluated the grown checkpoints *without continued training*. Under the hypothesis that proper continued training will not degrade performance, we believe the post-growth evaluation provides a reasonable proxy for assessing the effect of model growth on LR-fully-decayed models. Below, we report the results on four widely used open-source MoE models with comparable activated parameter sizes:
> >
> > | Model                        | Model Size  | Trained Tokens | MMLU (before) | QA Avg (before) | MMLU (after) | QA Avg (after) |
> > | ---------------------------- | ----------- | -------------- | ------------- | --------------- | ------------ | -------------- |
> > | deepseek-ai/deepseek-moe     | 16.4B_A2.8B | 2T             | 44.63         | 56.80         | 39.55        | 54.14        |
> > | deepseek-ai/Deepseek_V2_Lite | 16B_A2.4B   | 5.7T           | 58.04         | 58.77        | 55.49        | 57.12         |
> > | Qwen/Qwen1.5-MoE-A2.7B       | 14.3B_A2.7B | ~5T*           | 61.03         | 57.61        | 56.12        | 54.46        |
> > | Qwen/Qwen3-30B-A3B           | 30B_A3B     | 36T            | 79.50         | 62.21          | 75.69        | 61.07         |
> >
> > * *Qwen1.5-MoE-A2.7B is upcycled from Qwen1.8B, originally trained on 2.2T tokens. Since the upcycled training token count is not disclosed, we infer—based on its similarity to DeepSeek_V2_Lite—that it was likely trained on approximately 5T tokens in total.*
> >
> > From this table, we observe that the *direct post-growth performance* also follows a positive trend with respect to the sunk cost of the base converged model. With continued training, these grown checkpoints are expected to further improve, providing an additional, complementary confirmation that our proposed positive sunk-cost correlation extends to models trained with fully decayed LR schedules.
> >
> > ---
> >
> > We sincerely thank the reviewer once again for the valuable feedback. We would like to reiterate that although our practical growth procedures may appear straightforward, **our motivation that recycling highly converged MoE pretrained checkpoints with substantial sunk cost is both important and central to our work**. This perspective also explains why our findings differ from those reported in related studies.
> >
> > We have revised the manuscript in accordance with the reviewer’s suggestions to better highlight the contributions of our work. We greatly appreciate any further comments or feedback the reviewer may wish to share.

---

### Author Response · Authors · 2025-11-28

We sincerely thank all reviewers for their time and valuable feedback. We are encouraged that all reviewers recognized the impressive and practical experimental scale of our work. In this rebuttal, we systematically addressed all questions and concerns, including clarifications on related work, additional quantitative analyses, missing baselines, and several interesting directions for future research beyond the scope of this paper. All corresponding revisions and additional experiments have been incorporated into the updated manuscript and are highlighted in blue.

---

### Author Response · Authors · 2025-12-01
**Rebuttal Summary for the Updated Review Phase by Authors**

We thank all reviewers and ACs for their valuable time and thoughtful assessment of our work. We are aware that further discussion with reviewers was closed on Nov 28, 15:00 UTC, after we submitted our second-round rebuttal (Nov 28, 13:00 UTC). Therefore, we would like to summarize here the key questions raised by the reviewers and how our rebuttal addressed them.

---

- **For reviewer 6wRb (rating 2, confidence 3):**

This reviewer responded (Nov 22) to our first-round rebuttal (Nov 13) but did not have time to reply to our second-round rebuttal. We resolved the reviewer’s central concerns, which stemmed primarily from a **misunderstanding of why our proposed growth method works**. In the first-round rebuttal, the reviewer acknowledged the fundamental differences between our work and prior studies regarding **width growth**, and raised insightful questions about **depth growth**.

We appreciate these suggestions and added a detailed quantitative analysis identifying the boundary at which *interposition* becomes superior to *stack*, accompanied by additional evidence based on our observation of layer-wise weight norms. This strengthened the soundness and verifiability of our main finding:

**Once a stable layer-wise increasing weight-norm pattern emerges in a pretrained MoE model, "interposition" outperforms "stack" because it preserves this pattern, whereas "stack" disrupts it.**

We also addressed remaining concerns, including the ad-hoc choice of growth factor and experiments involving LR-fully-decayed checkpoints, and showed that neither affects our core claims.

We would like to reiterate that although our practical growth strategies may appear straightforward, **our motivation on recycling highly converged MoE pretrained checkpoints with substantial sunk cost is the central and critical part of our paper**. This viewpoint also explains why our conclusions differ from earlier works.

- **For reviewer wSPh (rating 6, confidence 4):**

This reviewer did not have the opportunity to respond to our first-round rebuttal (Nov 21). We thank the reviewer for recognizing the strength of our experiments on the "interposition > stack" finding and the careful study of growth timing, as well as acknowledging the practicality and scale of our experiments. The reviewer raised a similar question about the boundary of superiority for the interposition method as reviewer 6wRb, which we have addressed through quantitative analysis. The reviewer also requested two additional baseline experiments, which we have added both in the rebuttal and in the updated manuscript.

- **For reviewer eGL6 (rating 4, confidence 4):**

This reviewer also did not have the opportunity to respond to our first-round rebuttal (Nov 21). We thank the reviewer for recognizing the insightfulness of our width/depth comparison and our effort in carefully matching FLOPs. We addressed concerns regarding missing details and baselines for MoE models, showing that these factors do **not** influence our conclusions about width-growth models. We also clarified misunderstandings related to several figures and related-work comparisons, and added supplementary experiments in response to the reviewer’s questions. These additions do not alter the main claims of the paper.

---

We thank all reviewers again for their time and engagement, even though they were unable to participate further in the rebuttal phase. We understand that the updated review phase is an unpredictable event. Accordingly, we provide this summary to clarify the efforts we made during the rebuttal process and to assist the new AC in understanding how we thoroughly addressed the reviewers’ concerns.

---

### Meta-Review · Area_Chair_1m8U · 2026-01-02

**Summary:**

This paper addresses the critical issue of high computational costs associated with pretraining large language models (LLMs). The authors propose a framework to "recycle" existing, well-trained MoE checkpoints by expanding them into larger, more capable models, thereby leveraging the "sunk cost" of prior computation. The framework leverages interpositional layer copying for depth growth, and expert duplication for width growth. Authors showed that a 17B MoE model can be expanded successfully to a 70B model and yields better performance compared to models trained from scratch with the same compute budget, which indicates efficacy of proposed framework.

Reviewers found the paper is of following strength

* Study a highly relevant and practical problem: finding more compute-efficient ways to scale LLMs. This is crucial for the sustainability and accessibility of AI research.
* The experiments were conducted at practical scale (70B with 1T tokens)
* The paper select reasonable benchmarks to demonstrate the effectiveness of proposed method. The evaluation setup is strong and makes the results convincing.
* The paper is very clear and well-written.

**Reviewer Concerns:**

We also found the additional explanation from authors during rebuttal are helpful (e.g., comparing proposed work with previous studies, insights on the depth/width growth , more experiments with various baselines, settings, and compute budget). These are very helpful experiments and discussion, and definitely strengthen this work.

Although this submission is of reasonable quality and contribution to science community, the reviewers still found this work is at the borderline compared to a typical ICLR paper. We understand the computation intensity for conducting LLM experiments at practical scale. However, we still feel there are more experiments should be done to better demystify why proposed method works and when (e.g., some hyperparameters and experiment settings are selected arbitrarily). Since the underlying model is at a very large scale where human intuition can't play much role, comprehensive experiments are necessary to show the proposed work are beneficial and how applicable it is to different circumstances. This is why we start seeing very long technical reports from every big lab working on LLMs. Due to this hesitation, we found this paper is too early to be introduced to ICLR audience and we encourage the authors to resubmit to a later conference.

**Reviewer Scores:**

I would expect the reviewers may slightly increase scores during the rebuttal with the additional experiments and discussion. However, with the concerns discussed above in details about scale of experiments, there is no clear distinction that would likely convince reviewers to accept this paper.

---

### Decision · Program_Chairs · 2026-01-26

Reject